# FUNCTION-SPACE REGULARIZED RÉNYI DIVERGENCES

**Jeremiah Birrell[1], Yannis Pantazis[2], Paul Dupuis[3], Luc Rey-Bellet[1], Markos A. Katsoulakis[1]**

[1]University of Massachusetts, Amherst, [2]Foundation for Research & Technology - Hellas, [3]Brown University, {jbirrell, luc, markos}@umass.edu, pantazis@iacm.forth.gr, paul_dupuis@brown.edu

## ABSTRACT

We propose a new family of regularized Rényi divergences parametrized not only by the order $\alpha$ but also by a variational function space. These new objects are defined by taking the infimal convolution of the standard Rényi divergence with the integral probability metric (IPM) associated with the chosen function space. We derive a novel dual variational representation that can be used to construct numerically tractable divergence estimators. This representation avoids risk-sensitive terms and therefore exhibits lower variance, making it well-behaved when $\alpha > 1$; this addresses a notable weakness of prior approaches. We prove several properties of these new divergences, showing that they interpolate between the classical Rényi divergences and IPMs. We also study the $\alpha \to \infty$ limit, which leads to a regularized worst-case-regret and a new variational representation in the classical case. Moreover, we show that the proposed regularized Rényi divergences inherit features from IPMs such as the ability to compare distributions that are not absolutely continuous, e.g., empirical measures and distributions with low-dimensional support. We present numerical results on both synthetic and real datasets, showing the utility of these new divergences in both estimation and GAN training applications; in particular, we demonstrate significantly reduced variance and improved training performance.

## 1 INTRODUCTION

Rényi divergence, Rényi (1961), is a significant extension of Kullback-Leibler (KL) divergence for numerous applications; see, e.g., Van Erven & Harremos (2014). The recent neural-based estimators for divergences Belghazi et al. (2018) along with generative adversarial networks (GANs) Goodfellow et al. (2014) accelerated the use of divergences in the field of deep learning. The neural-based divergence estimators are feasible through the utilization of variational representation formulas. These formulas are essentially lower bounds (and, occasionally, upper bounds) which are approximated by tractable statistical averages. The estimation of a divergence based on variational formulas is a notoriously difficult problem. Challenges include potentially high bias that may require an exponential number of samples McAllester & Stratos (2020) or the exponential statistical variance for certain variational estimators Song & Ermon (2019), rendering divergence estimation both data inefficient and computationally expensive. This is especially prominent for Rényi divergences with order larger than 1. Indeed, numerical simulations have shown that, unless the distributions $P$ and $Q$ are very close to one another, the Rényi divergence $R_\alpha(P\|Q)$ is almost intractable to estimate when $\alpha > 1$ due to the high variance of the statistically-approximated risk-sensitive observables Birrell et al. (2021), see also the recent analysis in Lee & Shin (2022). A similar issue has also been observed for the KL divergence, Song & Ermon (2019). Overall, the lack of estimators with low variance for Rényi divergences has prevented wide-spread and accessible experimentation with this class of information-theoretic tools, except in very special cases. We hope our results here will provide a suitable set of tools to address this gap in the methodology.

One approach to variance reduction is the development of new variational formulas. This direction is especially fruitful for the estimation of mutual information van den Oord et al. (2018); Cheng et al. (2020). Another approach is to regularize the divergence by restricting the function space of the variational formula. Indeed, instead of directly attacking the variance issue, the function space of the variational formula can be restricted, for instance, by bounding the test functions or more appropriately by bounding the derivative of the test functions. The latter regularization leads to

Lipschitz continuous function spaces which are also foundational to integral probability metrics (IPMs) and more specifically to the duality property of the Wasserstein metric. In this paper we combine the above two approaches, first deriving a new variational representation of the classical Rényi divergences and then regularizing via an infimal-convolution as follows

$$R_\alpha^{\Gamma,IC}(P\|Q) := \inf_\eta\{R_\alpha(P\|\eta) + W^\Gamma(Q,\eta)\}, \tag{1}$$

where $P$ and $Q$ are the probability distributions being compared, the infimum is over the space of probability measures, $R_\alpha$ is the classical Rényi divergence, and $W^\Gamma$ is the IPM corresponding to the chosen regularizing function space, $\Gamma$.

The new family of regularized Rényi divergences that are developed here address the risk-sensitivity issue inherent in prior approaches. More specifically, our contributions are as follows.

- We define a new family of function-space regularized Rényi divergences via the infimal convolution operator between the classical Rényi divergence and an arbitrary IPM (1). The new regularized Rényi divergences inherit their function space from the IPM. For instance, they inherit mass transport properties when one regularizes using the 1-Wasserstein metric.

- We derive a dual variational representation (11) of the regularized Rényi divergences which avoids risk-sensitive terms and can therefore be used to construct lower-variance statistical estimators.

- We prove a series of properties for the new object: (a) the divergence property, (b) being bounded by the minimum of the Rényi divergence and IPM, thus allowing for the comparison of non-absolutely continuous distributions, (c) limits as $\alpha \to 1$ from both left and right, (d) regimes in which the limiting cases $R_\alpha(P\|Q)$ and $W^\Gamma(Q,P)$ are recovered.

- We propose a rescaled version of the regularized Rényi divergences (16) which lead to a new variational formula for the worst-case regret (i.e., $\alpha \to \infty$). This new variational formula does not involve the essential supremum of the density ratio as in the classical definition of worst-case regret, thereby avoiding risk-sensitive terms.

- We present a series of illustrative examples and counterexamples that further motivate the proposed definition for the function-space regularized Rényi divergences.

- We present numerical experiments that show (a) that we can estimate the new divergence for large values of the order $\alpha$ without variance issues and (b) train GANs using regularized function spaces.

**Related work.** The order of Rényi divergence controls the weight put on the tails, with the limiting cases being mode-covering and mode-selection Minka (2005). Rényi divergence estimation is used in a number of applications, including Sajid et al. (2022) (behavioural sciences), Mironov (2017) (differential privacy), and Li & Turner (2016) (variational inference); in the latter the variational formula is an adaptation of the evidence lower bound. Rényi divergences have been also applied in the training of GANs Bhatia et al. (2021) (loss function for binary classification - discrete case) and in Pantazis et al. (2022) (continuous case, based on the Rényi-Donsker-Varahdan variational formula in Birrell et al. (2021)). Rényi divergences with $\alpha > 1$ are also used in contrastive representation learning, Lee & Shin (2022), as well as in PAC-Bayesian Bounds, Bégin et al. (2016). In the context of uncertainty quantification and sensitivity analysis, Rényi divergences provide confidence bounds for rare events, Atar et al. (2015); Dupuis et al. (2020), with higher rarity corresponding to larger $\alpha$.

Reducing the variance of divergence estimators through control of the function space have been recently proposed. In Song & Ermon (2019) an explicit bound to the output restricts the divergence values. A systematic theoretical framework on how to regularize through the function space has been developed in Dupuis, Paul & Mao, Yixiang (2022); Birrell et al. (2022a) for the KL and $f$-divergences. Despite not covering the Rényi divergence, the theory in Dupuis, Paul & Mao, Yixiang (2022); Birrell et al. (2022a) and particularly the infimal-convolution formulation clearly inspired the current work. However, adapting the infimal-convolution method to the Rényi divergence setting requires two new technical innovations: (a) We develop a new low-variance convex-conjugate variational formula for the classical Rényi divergence in Theorem 2.1 (see also Fig. 1), allowing us to apply infimal-convolution tools to develop the new $\Gamma$-Rényi divergences in Theorem 3.4. (b) We study the $\alpha \to \infty$ limit of (a) to obtain a new low-variance variational representation of worst-case regret in Theorem 2.2 and study its $\Gamma$-regularization in Theorem 4.5.

## 2 NEW VARIATIONAL REPRESENTATIONS OF CLASSICAL RÉNYI DIVERGENCES

The Rényi divergence of order $\alpha \in (0,1) \cup (1,\infty)$ between $P$ and $Q$, denoted $R_\alpha(P\|Q)$, can be defined as follows: Let $\nu$ be a sigma-finite positive measure with $dP = pd\nu$ and $dQ = qd\nu$. Then

$$R_\alpha(P\|Q) := \begin{cases} \frac{1}{\alpha(\alpha-1)} \log\left[\int_{q>0} p^\alpha q^{1-\alpha} d\nu\right] & \text{if } 0 < \alpha < 1 \text{ or} \\ & \alpha > 1 \text{ and } P \ll Q \\ +\infty & \text{if } \alpha > 1 \text{ and } P \not\ll Q, \end{cases} \tag{2}$$

where $P \ll Q$ denotes absolute continuity of $P$ with respect to $Q$. There always exists such a $\nu$ (e.g., $\nu = P + Q$) and one can show that the definition (2) does not depend on the choice of $\nu$. The $R_\alpha$ provide a notion of 'distance' between $P$ and $Q$ in that they satisfy the divergence property, i.e., they are non-negative and equal zero iff $Q = P$. The limit of $R_\alpha$ as $\alpha$ approaches 1 or 0 equals the KL or reverse KL divergence respectively Van Erven & Harremos (2014).

An alternative representation of $R_\alpha$, the so-called Rényi-Donsker-Varadhan variational formula, was derived from (2) in Birrell et al. (2021),

$$R_\alpha(P\|Q) = \sup_{\phi \in \mathcal{M}_b(\Omega)} \left\{ \frac{1}{\alpha-1} \log \int e^{(\alpha-1)\phi} dP - \frac{1}{\alpha} \log \int e^{\alpha\phi} dQ \right\}, \quad P, Q \in \mathcal{P}(\Omega). \tag{3}$$

Here $(\Omega, \mathcal{M})$ denotes a measurable space, $\mathcal{M}_b(\Omega)$ the space of bounded measurable real-valued functions on $\Omega$, and $\mathcal{P}(\Omega)$ is the space of probability measures on $\Omega$. By a change of variables argument this can be transformed into the following new variational representation; see Theorem A.2 in Appendix A for a proof. We call it the **convex-conjugate Rényi variational formula** (**CC-Rényi**).

**Theorem 2.1** (Convex-Conjugate Rényi Variational Formula). *Let $P, Q \in \mathcal{P}(\Omega)$ and $\alpha \in (0,1) \cup (1,\infty)$. Then*

$$R_\alpha(P\|Q) = \sup_{g \in \mathcal{M}_b(\Omega):g<0} \left\{ \int g dQ + \frac{1}{\alpha-1} \log \int |g|^{(\alpha-1)/\alpha} dP \right\} + \alpha^{-1}(\log\alpha + 1). \tag{4}$$

*If $(\Omega, \mathcal{M})$ is a metric space with the Borel $\sigma$-algebra then (4) holds with $\mathcal{M}_b(\Omega)$ replaced by $C_b(\Omega)$, the space of bounded continuous real-valued functions on $\Omega$.*

The representation (4) is of convex-conjugate type, which will be key in our development of function-space regularized Rényi divergences. It is also of independent interest as it avoids risk-sensitive terms, unlike (3) which contains cumulant-generating-functions. This makes (4) better behaved in estimation problems, especially when $\alpha > 1$; see the example in Section 6.1 below.

We also obtain a new variational formula for worst-case regret, as defined by Van Erven & Harremos (2014)

$$D_\infty(P\|Q) := \lim_{\alpha \to \infty} \alpha R_\alpha(P\|Q) = \begin{cases} \log\left(\text{ess sup}_P \frac{dP}{dQ}\right), & P \ll Q \\ \infty, & P \not\ll Q. \end{cases} \tag{5}$$

In contrast to (5), which requires estimation of the likelihood ratio, the new variational formula (6) below avoids risk-sensitive terms.

**Theorem 2.2** (Worst-case Regret Variational Formula). *Let $P, Q \in \mathcal{P}(\Omega)$. Then*

$$D_\infty(P\|Q) = \sup_{g \in \mathcal{M}_b(\Omega):g<0} \left\{ \int g dQ + \log \int |g| dP \right\} + 1. \tag{6}$$

*If $\Omega$ is a metric space with the Borel $\sigma$-algebra then (6) holds with $\mathcal{M}_b(\Omega)$ replaced by $C_b(\Omega)$.*

See Theorem A.5 in Appendix A for a proof. Equation (6) is a new result of independent interest and will also be useful in our study of the $\alpha \to \infty$ limit of the function-space regularized Rényi divergences that we define in the next section.

**Remark 2.3.** *Alternative variational formulas for $D_\infty$ on a finite alphabet were derived in Kurri et al. (2022).*

t

## 3 PRIMAL AND DUAL FORMULATIONS OF THE INFIMAL-CONVOLUTION Γ-RÉNYI DIVERGENCES

We are now ready to define the function-space regularized Rényi divergences and derive their key properties. In this section, $X$ will denote a compact metric space, $\mathcal{P}(X)$ will denote the set of Borel probability measures on $X$, and $C(X)$ will denote the space of continuous real-valued functions on $X$. We equip $C(X)$ with the supremum norm and recall that the dual space of $C(X)$ is $C(X)^* = M(X)$, the space of finite signed Borel measures on $X$ (see the Riesz representation theorem, e.g., Theorem 7.17 in Folland (2013)).

**Definition 3.1.** *Given a test-function space $\Gamma \subset C(X)$, we define the* **infimal-convolution Γ-Rényi divergence** *(i.e.,* **IC-Γ-Rényi divergence**) *between $P, Q \in \mathcal{P}(X)$ by*

$$R_\alpha^{\Gamma,IC}(P\|Q) \coloneqq \inf_{\eta \in \mathcal{P}(X)} \{R_\alpha(P\|\eta) + W^\Gamma(Q,\eta)\}, \ \ \alpha \in (0,1) \cup (1,\infty), \tag{7}$$

*where $W^\Gamma$ denotes the Γ-IPM*

$$W^\Gamma(\mu,\nu) \coloneqq \sup_{g \in \Gamma}\{\int g d\mu - \int g d\nu\}, \ \ \mu, \nu \in M(X). \tag{8}$$

**Remark 3.2.** *The classical Rényi divergence is convex in its second argument but not in its first when $\alpha > 1$ Van Erven & Harremos (2014). This is the motivation for defining the IC-Γ-Rényi divergences via an infimal convolution in the second argument of $R_\alpha$; convex analysis tools will be critical in deriving properties of $R_\alpha^{\Gamma,IC}$ below. For $\alpha \in (0,1)$ one can use the identity $R_\alpha(P\|Q) = R_{1-\alpha}(Q\|P)$ to rewrite (7) as an infimal convolution in the first argument.*

The definition (7) can be thought of as a regularization of the classical Rényi divergence using the Γ-IPM. For computational purposes it is significantly more efficient to have a dual formulation, i.e., a representation of $R_\alpha^{\Gamma,IC}$ in terms of a supremum over a function space. To derive such a representation we begin with the variational formula for $R_\alpha$ from Theorem 2.1. If we define the convex mapping $\Lambda_\alpha^P : C(X) \to (-\infty, \infty]$,

$$\Lambda_\alpha^P[g] \coloneqq \infty 1_{g \not< 0} - \left(\frac{1}{\alpha-1}\log\int |g|^{(\alpha-1)/\alpha}dP + \alpha^{-1}(\log\alpha + 1)\right)1_{g<0}, \tag{9}$$

then (4) from Theorem 2.1 can be written as a convex conjugate

$$R_\alpha(P\|Q) = (\Lambda_\alpha^P)^*[Q] \coloneqq \sup_{g \in C(X)}\{\int g dQ - \Lambda_\alpha^P[g]\}. \tag{10}$$

One can then use Fenchel-Rockafellar duality to derive a dual formulation of the IC-Γ-Rényi divergences. To apply this theory we will need to work with spaces of test functions that satisfy the following admissibility properties. These properties are similar to those used in the construction of regularized KL and $f$-divergences in Dupuis, Paul & Mao, Yixiang (2022) and Birrell et al. (2022a).

**Definition 3.3.** *We will call $\Gamma \subset C(X)$ **admissible** if it is convex and contains the constant functions. We will call an admissible $\Gamma$ **strictly admissible** if there exists a $\mathcal{P}(X)$-determining set $\Psi \subset C(X)$ such that for all $\psi \in \Psi$ there exists $c \in \mathbb{R}$, $\epsilon > 0$ such that $c \pm \epsilon\psi \in \Gamma$. Recall that $\Psi$ being $\mathcal{P}(X)$-**determining** means that for all $Q, P \in \mathcal{P}(X)$, if $\int \psi dQ = \int \psi dP$ for all $\psi \in \Psi$ then $Q = P$.*

Putting the above pieces together one obtains the following variational representation.

**Theorem 3.4.** *Let $\Gamma \subset C(X)$ be admissible, $P, Q \in \mathcal{P}(X)$, and $\alpha \in (0,1) \cup (1,\infty)$. Then:*

*1.*

$$R_\alpha^{\Gamma,IC}(P\|Q) = \sup_{g \in \Gamma : g < 0}\left\{\int g dQ + \frac{1}{\alpha-1}\log\int |g|^{(\alpha-1)/\alpha}dP\right\} + \alpha^{-1}(\log\alpha + 1). \tag{11}$$

*2. If (11) is finite then there exists $\eta_* \in \mathcal{P}(X)$ such that*

$$R_\alpha^{\Gamma,IC}(P\|Q) = \inf_{\eta \in \mathcal{P}(X)} \{R_\alpha(P\|\eta) + W^\Gamma(Q,\eta)\} = R_\alpha(P\|\eta_*) + W^\Gamma(Q,\eta_*). \tag{12}$$

3. $R_\alpha^{\Gamma,IC}(P\|Q) \leq \min\{R_\alpha(P\|Q), W^\Gamma(Q,P)\}$.

4. *If $\Gamma$ is strictly admissible then $R_\alpha^{\Gamma,IC}$ has the divergence property.*

See Theorem B.3 in Appendix B for detailed proofs of these results as well as several additional properties. We note that there are alternative strategies for proving the variational formula (11) which make different assumptions; further comments on this can be found in Remark B.4. Important examples of strictly admissible $\Gamma$ include the following:

1. $\Gamma = C(X)$, which leads to the classical Rényi-divergences.

2. $\Gamma = \text{Lip}^1(X)$, i.e. all 1-Lipschitz functions. This regularizes the Rényi divergences via the Wasserstein metric.

3. $\Gamma = \{c + g : c \in \mathbb{R}, g \in C(X), |g| \leq 1\}$. This regularizes the Rényi divergences via the total-variation metric.

4. $\Gamma = \{c + g : c \in \mathbb{R}, g \in \text{Lip}^1(X), |g| \leq 1\}$. This regularizes the Rényi divergences via the Dudley metric.

5. $\Gamma = \{c + g : c \in \mathbb{R}, g \in Y : \|g\|_V \leq 1\}$, the unit ball in a RKHS $V \subset C(X)$. This regularizes the Rényi divergences via MMD.

In practice, uniform bounds can be implemented using an appropriately chosen final NN layer. Lipschitz bounds can be implemented using spectral normalization of neural networks Miyato et al. (2018), or using a soft gradient penalty Gulrajani et al. (2017). The function space $\Gamma$ for structure-preserving GANs discussed in the Appendix is implemented using equivariant neural networks, Birrell et al. (2022b). If $\Gamma$ is a ball in an RKHS space the implementation is carried out using the same tools used in, e.g., MMD distances and divergences, Gretton et al. (2012); Glaser et al. (2021).

The IC-$\Gamma$-Rényi divergences also satisfy a data processing inequality. See Theorem B.8 in Appendix B for a proof as well as details regarding the notation.

**Theorem 3.5** (Data Processing Inequality). *Let $\alpha \in (0,1) \cup (1,\infty)$, $Q, P \in \mathcal{P}(X)$, and $K$ be a probability kernel from $X$ to $Y$ such that $K[g] \in C(X)$ for all $g \in C(X,Y)$. If $\Gamma \subset C(Y)$ is admissible then $R_\alpha^{\Gamma,IC}(K[P]\|K[Q]) \leq R_\alpha^{K[\Gamma],IC}(P\|Q)$. If $\Gamma \subset C(X \times Y)$ is admissible then $R_\alpha^{\Gamma,IC}(P \otimes K\|Q \otimes K) \leq R_\alpha^{K[\Gamma],IC}(P\|Q)$.*

If $K[\Gamma]$ is strictly contained in $\Gamma$ then the bounds in Theorem 3.5 can be strictly tighter than the classical data processing inequality Van Erven & Harremos (2014). Data-processing inequalities are important for constructing symmetry-preserving GANs; see Birrell et al. (2022b) and Section D.1.

## 4 LIMITS, INTERPOLATIONS, AND REGULARIZED WORST-CASE REGRET

Next we use Theorem 3.4 to compute various limits of the IC-$\Gamma$-Rényi divergences. First we show that they interpolate between $R_\alpha$ and $W^\Gamma$ in the following sense (see Theorem B.5 for a proof).

**Theorem 4.1.** *Let $\Gamma \subset C(X)$ be admissible, $P, Q \in \mathcal{P}(X)$, and $\alpha \in (0,1) \cup (1,\infty)$.*

1. $\lim_{\delta \to 0^+} \frac{1}{\delta} R_\alpha^{\delta\Gamma,IC}(P\|Q) = W^\Gamma(Q,P)$,

2. *If $\Gamma$ is strictly admissible then $\lim_{L \to \infty} R_\alpha^{L\Gamma,IC}(P\|Q) = R_\alpha(P\|Q)$.*

Now we discuss the limiting behavior in $\alpha$. These results generalize several properties of the classical Rényi divergences Van Erven & Harremos (2014). First we consider the $\alpha \to 1$ limit; see Theorem B.6 for a proof.

**Theorem 4.2.** *Let $\Gamma \subset C(X)$ be admissible and $P, Q \in \mathcal{P}(X)$. Then*

$$\lim_{\alpha \to 1^+} R_\alpha^{\Gamma,IC}(P\|Q) = \inf_{\substack{\eta \in \mathcal{P}(X): \\ \exists \beta > 1, R_\beta(P\|\eta) < \infty}} \{R(P\|\eta) + W^\Gamma(Q,\eta)\}, \tag{13}$$

$$\lim_{\alpha \to 1^-} R_\alpha^{\Gamma,IC}(P\|Q) = \inf_{\eta \in \mathcal{P}(X)} \{R(P\|\eta) + W^\Gamma(Q,\eta)\} \tag{14}$$

$$= \sup_{g \in \Gamma: g < 0} \left\{\int g dQ + \int \log|g| dP\right\} + 1. \tag{15}$$

**Remark 4.3.** *When $\Gamma = C(X)$, changing variables to $g = -\exp(\phi - 1)$ transforms (15) into the Legendre-transform variational formula for $R(P\|Q)$; see equation (1) in Birrell et al. (2022c) with $f(x) = x\log(x)$. Eq. (14) is an infimal convolution of the reverse KL-divergence, as opposed to the results in Dupuis, Paul & Mao, Yixiang (2022) which apply to the (forward) KL-divergence.*

**Function-space regularized worst-case regret.** Next we investigate the $\alpha \to \infty$ limit of the IC-$\Gamma$-Rényi divergences, which will lead to the function-space regularized worst-case regret. First recall that some authors use an alternative definition of the classical Rényi divergences, related to the one used in this paper by $D_\alpha(\cdot\|\cdot) \coloneqq \alpha R_\alpha(\cdot\|\cdot)$. This alternative definition has the useful property of being non-decreasing in $\alpha$; see Van Erven & Harremos (2014). Appropriately rescaled, the IC-$\Gamma$-Rényi divergence also satisfies this property, leading to the following definition.

**Definition 4.4.** *For $\Gamma \subset C(X)$, $\alpha \in (0,1) \cup (1,\infty)$ and $P, Q \in \mathcal{P}(X)$ we define*

$$D_\alpha^{\Gamma,IC}(P\|Q) \coloneqq \alpha R_\alpha^{\Gamma/\alpha,IC}(P\|Q). \tag{16}$$

Note that $\alpha R_\alpha^{\Gamma/\alpha,IC}(P\|Q)$ is non-decreasing in $\alpha$; see Lemma B.1 for a proof. We now show that the divergences $D_\alpha^{\Gamma,IC}$ are well behaved in the $\alpha \to \infty$ limit, generalizing (5). Taking this limit provides a definition of function-space regularized worst-case regret, along with the following dual variational representation.

**Theorem 4.5.** *Let $\Gamma \subset C(X)$ be admissible and $P, Q \in \mathcal{P}(X)$. Then*

$$D_\infty^{\Gamma,IC}(P\|Q) \coloneqq \lim_{\alpha\to\infty} D_\alpha^{\Gamma,IC}(P\|Q) = \inf_{\eta\in P(X)} \left\{ D_\infty(P\|\eta) + W^\Gamma(Q,\eta) \right\} \tag{17}$$

$$= \sup_{g\in\Gamma:g<0} \left\{ \int g\,dQ + \log \int |g|\,dP \right\} + 1. \tag{18}$$

We call $D_\infty^{\Gamma,IC}$ the **infimal-convolution $\Gamma$-worst-case regret** (i.e., **IC-$\Gamma$-WCR**). The method of proof of Theorem 4.5 is similar to that of part (1) of Theorem 3.4; see Theorem B.7 in Appendix B for details. Theorem 4.5 suggests that $D_\alpha^{\Gamma,IC}$ is the appropriate $\alpha$-scaling to use when $\alpha$ is large and we find this to be the case in practice; see the example in Section 6.3.1.

## 5 ANALYTICAL EXAMPLES AND COUNTEREXAMPLES

In this section we present several analytical examples and counterexamples that illustrate important properties of the IC-$\Gamma$-Rényi divergences and demonstrate weaknesses of other attempts to define regularized Rényi divergences. In particular, we show that other attempts at regularizing Rényi divergences fail to inherit important properties from the $\Gamma$-IPM. More details on the computations can be found in Appendix C

**Infimal convolution and scaling limits:** First we present a simple example that illustrates the infimal convolution formula and limiting properties from Sections 3 and 4. Let $P = \delta_0$, $Q_{x,c} = c\delta_0 + (1-c)\delta_x$ for $c \in (0,1)$, $x > 0$, and let $\Gamma = \mathrm{Lip}^1$. Then for $L > 0$ one can compute

$$R_\alpha^{L\Gamma,IC}(P\|Q_{x,c}) = \begin{cases} (1-c)Lx\,, & 0 < \alpha Lx < 1 \\ \alpha^{-1} - cLx + \alpha^{-1}\log(\alpha Lx)\,, & 1 \le \alpha Lx \le 1/c \\ \alpha^{-1}\log(1/c)\,, & \alpha Lx > 1/c \end{cases} \tag{19}$$

In particular, it is straightforward to show that $R_\alpha^{L\Gamma,IC}(P\|Q_{x,c}) \le (1-c)Lx = W^{L\Gamma}(Q_{x,c},P)$, $\lim_{x\to 0^+} R_\alpha^{L\Gamma,IC}(P\|Q_{x,c}) = \lim_{x\to 0^+}(1-c)Lx = 0$, and $\lim_{L\to\infty} R_\alpha^{L\Gamma,IC}(P\|Q_{x,c}) = \alpha\log(1/c) = R_\alpha(P\|Q_{x,c})$. We can also rewrite this in terms of the solution to the infimal convolution problem and take the worst-case-regret scaling limit as follows

$$R_\alpha^{L\Gamma,IC}(P\|Q_{x,c}) = \begin{cases} W^{L\Gamma}(Q_{x,c},P)\,, & 0 < \alpha Lx < 1 \\ R_\alpha(P\|Q_{x,1/(\alpha Lx)}) + W^{L\Gamma}(Q_{x,c},Q_{x,1/(\alpha Lx)})\,, & 1 \le \alpha Lx \le 1/c \\ R_\alpha(P\|Q_{x,c})\,, & \alpha Lx > 1/c \end{cases},$$

$$\lim_{\alpha\to\infty} \alpha R_\alpha^{\Gamma/\alpha,IC}(P\|Q_{x,c}) = \begin{cases} W^\Gamma(Q_{x,c},P)\,, & 0 < x < 1 \\ D_\infty(P\|Q_{x,1/x}) + W^\Gamma(Q_{x,c},Q_{x,1/x})\,, & 1 \le x \le 1/c \\ D_\infty(P\|Q_{x,c})\,, & x > 1/c \end{cases}. \tag{20}$$

$\Gamma$-**Rényi-Donsker-Varadhan counterexample:** As an alternative to Definition 3.1, one can attempt to regularize the Rényi divergences by restricting the test-function space in the variational representation (3), leading to the $\Gamma$-**Rényi-Donsker-Varadhan** ($\Gamma$-**Rényi-DV**) divergences

$$R_\alpha^{\Gamma,DV}(P\|Q) \coloneqq \sup_{\phi\in\Gamma}\left\{\frac{1}{\alpha-1}\log\int e^{(\alpha-1)\phi}dP - \frac{1}{\alpha}\log\int e^{\alpha\phi}dQ\right\}. \tag{21}$$

The bound $\log\int e^{c\phi}dP \geq c\int\phi dP$ for all $\phi\in\Gamma$, $c\in\mathbb{R}$ implies that $R_\alpha^{\Gamma,DV}\leq W^\Gamma$ for $\alpha\in(0,1)$, making (21) a useful regularization of the Rényi divergences in this case; this utility was demonstrated in Pantazis et al. (2022), where it was used to construct GANs. However, estimators built from the representation (21) (i.e., replacing $P$ and $Q$ by empirical measures) are known to be numerically unstable when $\alpha > 1$. Below we provide a counterexample showing that, unlike for the IC-$\Gamma$-Rényi divergences, $R_\alpha^{\Gamma,DV}\not\leq W^\Gamma$ in general when $\alpha > 1$. We conjecture that this is a key reason for the instability of $\Gamma$-Rényi-Donsker-Varadhan estimators when $\alpha > 1$.

Let $P_{x,c} = c\delta_0 + (1-c)\delta_x$, $Q = \delta_0$ for $x > 0$, $c \in (0,1)$ and $\Gamma_L = \mathrm{Lip}^L$. Then for $\alpha > 1$ we have $R_\alpha^{\Gamma_L,DV}(P_{x,c}\|Q) = \frac{1}{\alpha-1}\log\left(c + (1-c)\exp((\alpha-1)Lx)\right)$ and $W^{\Gamma_L}(P_{x,c},Q) = (1-c)Lx$. Using strict concavity of the logarithm one can then obtain the bound

$$R_\alpha^{\Gamma_L,DV}(P_{x,c}\|Q) > W^{\Gamma_L}(P_{x,c},Q). \tag{22}$$

This shows that, when $\alpha > 1$, $\Gamma$-Rényi-DV violates the key property that allows the IC-$\Gamma$-Rényi divergences to inherit properties from the corresponding $\Gamma$-IPM. Another alternative is to begin with (3) and then reduce the test-function space to $\frac{1}{\alpha}\log(\Gamma)$, where the logarithm is introduced to eliminate the exponential functions in (21). However, this definition also fails to provide an appropriate regularized Rényi divergence; in particular, it is incapable of meaningfully comparing Dirac distributions. See Appendix C.3 for details. These counterexamples lend further credence to our infimal-convolution based regularization approach (7).

## 6 NUMERICAL EXPERIMENTS

In this section we present numerical examples that demonstrate the use of the IC-$\Gamma$-Rényi divergences for both estimation and training of GANs (additional examples can be found in Appendix D). All of the divergences considered in this paper have a variational representation of the form $D(P\|Q) = \sup_{g\in\Gamma} H[g;P,Q]$ for some objective functional $H$; we use the corresponding estimator

$$\widehat{D}_n(P\|Q) \coloneqq \sup_{\theta\in\Theta} H[g_\theta; P_n, Q_n] \tag{23}$$

where $P_n$, $Q_n$ are $n$-sample empirical measures and $g_\theta$ is a family of neural networks (NN) with parameters $\theta\in\Theta$. For Lipschitz function spaces we weaken the Lipschitz constraint to a soft 1-sided gradient penalty (see Section 4.1 of Birrell et al. (2022a)). Optimization is performed using the Adam optimizer Kingma & Ba (2014). For the infimal convolution divergences we enforce negativity of the test function (i.e., discriminators) using a final layer having one of the following forms: 1)$-abs(x)$ or 2) $-(1/(1-x)1_{x<0} + (1+x)1_{x\geq 0})$. The latter, which we term poly-softplus, is $C^1$ and decays like $O(x^{-1})$ as $x\to-\infty$.

### 6.1 VARIANCE OF RÉNYI ESTIMATORS

As a first example, we compare estimators of the classical Rényi divergences (i.e., without regularization) constructed from DV-Rényi (3) and CC-Rényi (4) in a simple case where the exact Rényi divergence is known. We let $Q$ and $P$ be 1000-dimensional Gaussians with equal variance and study $R_\alpha(P\|Q)$ as a function of the separation between their means. The results are shown in Figure 1. We see that the estimator based on the convex-conjugate Rényi variational formula 4 has smaller variance and mean-squared error (MSE) that the Rényi-Donsker-Varadhan variational formula 3, with the difference becoming very large when $\alpha \gg 1$ or when $P$ and $Q$ are far apart (i.e., when $\mu_q$ is large). The Rényi-Donsker-Varadhan estimator only works well when $\mu_q$ and $\alpha$ are both not too large, but even in such cases the convex-conjugate Rényi estimator generally performs better. We conjecture that this difference is due to the presence of risk-sensitive terms in (3) which were eliminated in the new representation (4). We note that the NN for the convex-conjugate Rényi estimator used the poly-softplus final layer, as we found the $-abs$ final layer to result in a significant percentage of failed runs (i.e., NaN outputs) but this issue did not arise when using poly-softplus. We do not show results for either DV-WCR or CC-WCR here as the exact divergence is infinite in this example.

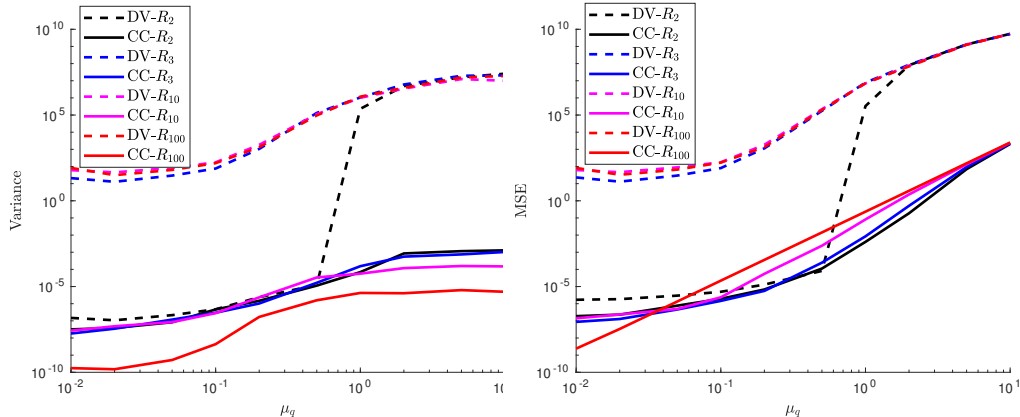

Figure 1: Variance and MSE of estimators of the classical Rényi divergence between 1000-dimensional Gaussians. DV-$R_\alpha$ refers to Rényi divergence estimators built using (3) while CC-$R_\alpha$ refers to estimators built using our new variational representation (4). We used a NN with one fully connected layer of 64 nodes, ReLU activations, and a poly-softplus final layer (for CC-Rényi). We trained for 10000 epochs with a minibatch size of 500. The variance and MSE were computing using data from 50 independent runs. Note that the CC-Rényi estimator has significantly reduced variance and MSE compared to the DV-Rényi estimator, even when $\alpha$ is large. Strikingly, the 1-D case exhibits the same behavior (see Figure 3 in Appendix D.2), demonstrating that the DV-Rényi estimator is unsuitable even in low dimensions.

## 6.2 DETECTION OF RARE SUB-POPULATIONS IN SINGLE-CELL BIOLOGICAL DATASETS

A critical task in cancer assessment is the detection of rare sub-populations subsumed in the overall population of cells. The advent of affordable flow and mass cytometry technologies that perform single cell measurements opens a new direction for the analysis and comparison of high-dimensional cell distributions Shahi et al. (2017) via divergence estimation. We consider single cell mass cytometry measurements on 16 bone marrow protein markers ($d = 16$) coming from healthy and disease individuals with acute myeloid leukemia Levine et al. (2015). Following Weber et al. (2019), we create two datasets: one with only healthy samples and another one with decreasing percentage of sick cells and compute several divergences. Considering the estimated divergence value as the score of a binary classifier, we compute the ROC curve and the respective area under the ROC curve (AUC) for any pair of sample distributions. More specifically, true negatives correspond to the divergence values between two healthy datasets while true positives correspond to the divergence between a healthy and a diseased dataset. Thus, the AUC is 1.0 when the divergence estimates are completely separable while AUC is 0.5 when they completely overlap. Table 1 reports the AUC values for the scaled IC-Γ-Rényi divergences (16), various levels of rarity and two sample sizes for the datasets. The best performance in the Rényi family is obtained for $\alpha = \infty$ using the IC-Γ-WCR variational formula (18). IC-Γ-WCR also outperforms the Wasserstein distance of first order in both sample size regimes.

## 6.3 IC-Γ-RÉNYI GANS

Finally, we study a pair of GAN examples (the second example is presented in Appendix D). Here the goal is to learn a distribution $P$ using a family of generator distribution $Q_\psi \sim h_\psi(X)$ where $X$ is a noise source and $h_\psi$ is a family of neural networks parametrized by $\psi \in \Psi$, i.e., the goal is to solve

$$\inf_{\psi \in \Psi} \widehat{D}_n(P \| Q_\psi)\,, \tag{24}$$

where $\widehat{D}_n$ is a divergence estimator of the form (23). In particular, we will study the GANs constructed from the newly introduced IC-Γ-Rényi and IC-Γ-WCR GANs and compare them with Wasserstein GAN Gulrajani et al. (2017); Arjovsky et al. (2017).

Table 1: AUC values (higher is better) for several divergences and various levels of rarity. The AUC values have been averaged from 50 independent runs. The neural discriminator has 2 hidden layers with 32 units each and ReLU activation. The $D_\alpha^{\Gamma,IC}$ divergences used the poly-softplus final layer.

| Sample size | | 100K | | | | | | | 20K | | | | |
|---|---|---|---|---|---|---|---|---|---|---|---|---|---|
| Probability (%) | | 0.1 | 0.2 | 0.3 | 0.4 | 0.5 | 1.0 | 10.0 | 0.1 | 0.3 | 0.5 | 1.0 | 10.0 |
| $D_\alpha^{\Gamma,IC}$ | $\alpha = 2$ | 0.51 | 0.55 | 0.62 | 0.64 | 0.70 | 0.92 | 1.00 | 0.48 | 0.58 | 0.58 | 0.60 | 1.00 |
| | $\alpha = 5$ | 0.66 | 0.70 | 0.71 | 0.72 | 0.74 | 0.80 | 1.00 | 0.32 | 0.37 | 0.43 | 0.38 | 0.91 |
| | $\alpha = 10$ | 0.57 | 0.50 | 0.62 | 0.64 | 0.49 | 0.59 | 1.00 | 0.48 | 0.48 | 0.43 | 0.47 | 0.74 |
| | $\alpha = \infty$ | 0.64 | 0.89 | 0.96 | 0.99 | 1.00 | 1.00 | 1.00 | 0.58 | 0.71 | 0.79 | 0.91 | 1.00 |
| Wasserstein | | 0.63 | 0.58 | 0.58 | 0.51 | 0.57 | 0.55 | 1.00 | 0.46 | 0.40 | 0.45 | 0.40 | 1.00 |

### 6.3.1 CIFAR-10

In Figure 2 we demonstrate improved performance of the IC-$\Gamma$-Rényi and IC-$\Gamma$-WCR GANs, as compared to Wasserstein GAN with gradient penalty (WGAN-GP), on the CIFAR-10 dataset Krizhevsky et al. (2009). The IC GANs also outperform Rényi-DV GAN (21), as the latter is highly unstable when $\alpha > 1$ and so the training generally encounters NaN after a small number of training epochs (hence we omit those results from the figure). We use the same ResNet neural network architecture as in (Gulrajani et al., 2017, Appendix F) and focus on evaluating the effect of different divergences. Here we let $\Gamma$ be the set of 1-Lipschitz functions, implement via a gradient penalty. Note that $D_\infty^{\Gamma,IC}$ performs significantly better than $R_\alpha^{\Gamma,IC}$ with large $\alpha$, and the rescaled $D_\alpha^{\Gamma,IC}$-GAN performs better that $R_\alpha^{\Gamma,IC}$-GAN when $\alpha$ is large.

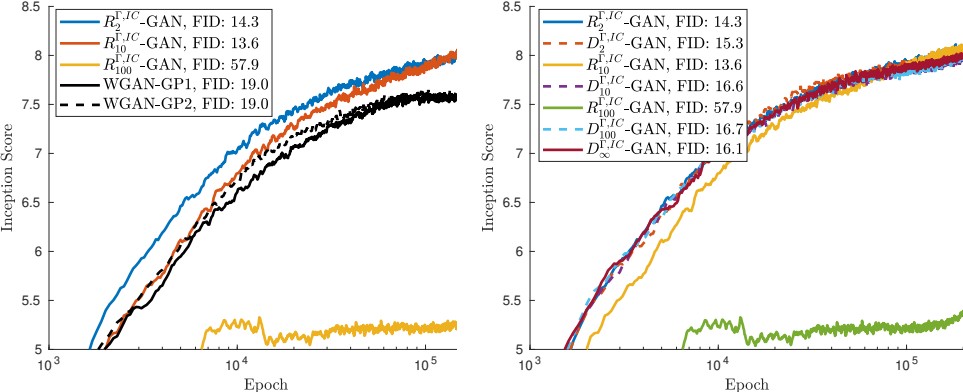

Figure 2: Comparison between IC-$\Gamma$-Rényi GAN, IC-$\Gamma$-WCR GAN, and WGAN-GP (both 1 and 2-sided) on the CIFAR-10 dataset. Here we plot the inception score as a function of the number of training epochs (moving average over the last 5 data points, with results averaged over 5 runs). We also show the averaged final FID score in the legend, computed using 50000 samples from both $P$ and $Q$. For the IC GANs we enforce negativity of the discriminator by using a final layer equal to $-$abs. The GANs were trained using the Adam optimizer with an initial learning rate of $0.0002$. The left pane shows that the IC-$\Gamma$-Rényi GANs outperform WGAN while the right pane shows that GANs based on the rescaled $D_\alpha^{\Gamma,IC}$ divergences (16) perform better when $\alpha$ is large, including in the $\alpha \to \infty$ limit, i.e., IC-$\Gamma$-WCR (17). In both cases the IC GANs outperform the $\Gamma$-Rényi-DV GANs with $\alpha > 1$ (21); the latter fail to converge due to the presence of risk-sensitive terms.

## ACKNOWLEDGMENTS

The research of J.B., M.K. and L.R.-B. was partially supported by the Air Force Office of Scientific Research (AFOSR) under the grant FA9550-21-1-0354. The research of M. K. and L.R.-B. was partially supported by the National Science Foundation (NSF) under the grants DMS-2008970 and TRIPODS CISE-1934846. The research of P.D. was partially supported by the NSF under the grant DMS-1904992 and by the AFOSR under the grant FA-9550-21-1-0354. The work of Y.P. was partially supported by the Hellenic Foundation for Research and Innovation (HFRI) through the "Second Call for HFRI Research Projects to support Faculty Members and Researchers" under Project 4753. This work was performed in part using high performance computing equipment obtained under a grant from the Collaborative R&D Fund managed by the Massachusetts Technology Collaborative.

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

## A   DERIVATION OF VARIATIONAL FORMULAS FOR THE CLASSICAL RÉNYI DIVERGENCES

In this appendix we provide several variational formulas for the classical Rényi divergences, some of which are new. In the following we let $(\Omega, \mathcal{M})$ denote a measurable space, $\mathcal{M}(\Omega)$ be the space of measurable real-valued functions on $\Omega$, $\mathcal{M}_b(\Omega)$ the subspace of bounded functions, and $\mathcal{P}(\Omega)$ will be the space of probability measures on $\Omega$.

First we recall the Rényi-Donsker-Varadhan variational formula derived in Birrell et al. (2021). This is a generalization of the Donsker-Varadhan variational representation of the KL divergence.

**Theorem A.1** (Rényi-Donsker-Varadhan Variational Formula). *Let $P$ and $Q$ be probability measures on $(\Omega, \mathcal{M})$ and $\alpha \in \mathbb{R}$, $\alpha \neq 0, 1$. Then for any set of functions, $\Phi$, with $\mathcal{M}_b(\Omega) \subset \Phi \subset \mathcal{M}(\Omega)$ we have*

$$R_\alpha(P\|Q) = \sup_{\phi \in \Phi} \left\{ \frac{1}{\alpha - 1} \log \int e^{(\alpha-1)\phi} dP - \frac{1}{\alpha} \log \int e^{\alpha\phi} dQ \right\}, \qquad (25)$$

*where we interpret $\infty - \infty \equiv -\infty$ and $-\infty + \infty \equiv -\infty$.*

*If in addition $(\Omega, \mathcal{M})$ is a metric space with the Borel $\sigma$-algebra then (25) holds for all $\Phi$ that satisfy $Lip_b(\Omega) \subset \Phi \subset \mathcal{M}(\Omega)$, where $Lip_b(\Omega)$ is the space of bounded Lipschitz functions on $\Omega$ (i.e., Lipschitz for any Lipschitz constant $L \in (0, \infty)$).*

Using Theorem A.1 we can derive a new variational representation that takes the form of a convex conjugate.

**Theorem A.2** (Convex-Conjugate Rényi Variational Formula). *Let $P, Q \in \mathcal{P}(\Omega)$ and $\alpha \in (0, 1) \cup (1, \infty)$. Then*

$$R_\alpha(P\|Q) = \sup_{g \in \mathcal{M}_b(\Omega): g < 0} \left\{ \int g dQ + \frac{1}{\alpha - 1} \log \int |g|^{(\alpha-1)/\alpha} dP \right\} + \alpha^{-1}(\log \alpha + 1). \quad (26)$$

*If $(\Omega, \mathcal{M})$ is a metric space with the Borel $\sigma$-algebra then (26) holds with $\mathcal{M}_b(\Omega)$ replaced by $C_b(\Omega)$, the space of bounded continuous real-valued functions on $\Omega$.*

*Proof.* Let $\Phi = \{\alpha^{-1}\log(-h) : h \in \mathcal{M}_b(\Omega), h < 0\}$. We have $\mathcal{M}_b(\Omega) \subset \Phi \subset \mathcal{M}(\Omega)$, hence Theorem A.1 implies

$$R_\alpha(P\|Q) = \sup_{\phi \in \Phi} \left\{ \frac{1}{\alpha - 1} \log \int e^{(\alpha-1)\phi} dP - \frac{1}{\alpha} \log \int e^{\alpha\phi} dQ \right\} \qquad (27)$$

$$= \sup_{h \in \mathcal{M}_b(\Omega): h < 0} \left\{ \frac{1}{\alpha - 1} \log \int e^{(\alpha-1)(\alpha^{-1}\log(-h))} dP - \frac{1}{\alpha} \log \int e^{\alpha(\alpha^{-1}\log(-h))} dQ \right\}$$

$$= \sup_{h \in \mathcal{M}_b(\Omega): h < 0} \left\{ \frac{1}{\alpha - 1} \log \int |h|^{(\alpha-1)/\alpha} dP - \frac{1}{\alpha} \log \int (-h) dQ \right\}.$$

Note that the second term is finite but the first term is possibly infinite when $\alpha \in (0, 1)$. Next use the identity

$$\log(c) = \inf_{z \in \mathbb{R}} \{z - 1 + ce^{-z}\}, \quad c \in (0, \infty) \qquad (28)$$

in the second term to write

$$R_\alpha(P\|Q) = \sup_{h \in \mathcal{M}_b(\Omega): h < 0} \left\{ \frac{1}{\alpha - 1} \log \int |h|^{(\alpha-1)/\alpha} dP - \frac{1}{\alpha} \inf_{z \in \mathbb{R}} \{z - 1 + e^{-z} \int (-h) dQ\} \right\} \tag{29}$$

$$= \sup_{z \in \mathbb{R}} \sup_{h \in \mathcal{M}_b(\Omega): h < 0} \left\{ \frac{1}{\alpha - 1} \log \int |h|^{(\alpha-1)/\alpha} dP + \frac{z - 1}{\alpha} + \alpha^{-1} e^{-z} \int h dQ \right\}.$$

For each $z \in \mathbb{R}$ make the change variables $h = \alpha e^z g$, $g \in \mathcal{M}_b(\Omega)$, $g < 0$ in the inner supremum to derive

$$R_\alpha(P\|Q) = \sup_{z \in \mathbb{R}} \sup_{g \in \mathcal{M}_b(\Omega): g < 0} \left\{ \frac{1}{\alpha - 1} \log \int |\alpha e^z g|^{(\alpha-1)/\alpha} dP - \frac{z - 1}{\alpha} + \alpha^{-1} e^{-z} \int \alpha e^z g dQ \right\} \tag{30}$$

$$= \sup_{z \in \mathbb{R}} \sup_{g \in \mathcal{M}_b(\Omega): g < 0} \left\{ \frac{1}{\alpha - 1} \log \int |g|^{(\alpha-1)/\alpha} dP + (\alpha^{-1}(\log \alpha + 1) + \int g dQ) \right\}$$

$$= \sup_{g \in \mathcal{M}_b(\Omega): g < 0} \left\{ \int g dQ + \frac{1}{\alpha - 1} \log \int |g|^{(\alpha-1)/\alpha} dP \right\} + \alpha^{-1}(\log \alpha + 1).$$

This completes the proof of (26). The proof of the metric-space version in nearly identical. $\qquad \square$

**Remark A.3.** *To reverse the above derivation and obtain (25) (with $\Phi = \{\phi \in \mathcal{M}(\Omega) : \phi \text{ is bounded above}\}$) from (26), change variables $g \mapsto -c \exp(\alpha\phi)$, $\phi \in \Phi$, $c > 0$ in (26) and then maximize over $c$.*

**Corollary A.4.** *If $X$ is a compact metric space with the Borel $\sigma$-algebra, $P, Q \in \mathcal{P}(X)$, and $\alpha \in (0,1) \cup (1,\infty)$ then $C_b(X) = C(X)$ and so*

$$R_\alpha(P\|Q) = \sup_{g \in C(X):g<0} \left\{ \int g dQ + \frac{1}{\alpha-1} \log \int |g|^{(\alpha-1)/\alpha} dP \right\} + \alpha^{-1}(\log\alpha + 1). \quad (31)$$

Next we derive a variational formula for the worst case regret, defined by

$$D_\infty(P\|Q) := \lim_{\alpha\to\infty} \alpha R_\alpha(P\|Q). \quad (32)$$

**Theorem A.5.** *Let $P, Q \in \mathcal{P}(\Omega)$. Then*

$$D_\infty(P\|Q) = \sup_{g \in \mathcal{M}_b(\Omega):g<0} \left\{ \int g dQ + \log \int |g| dP \right\} + 1. \quad (33)$$

*If $\Omega$ is a metric space with the Borel $\sigma$-algebra then (33) holds with $\mathcal{M}_b(\Omega)$ replaced by $C_b(\Omega)$.*

**Remark A.6.** *Note that on a compact metric space, the space of bounded continuous functions is the same as the space of all continuous functions.*

*Proof.* Recall Van Erven & Harremos (2014)

$$D_\infty(P\|Q) = \begin{cases} \log\left(\operatorname{ess\,sup}_P \frac{dP}{dQ}\right), & P \ll Q \\ \infty, & P \not\ll Q. \end{cases} \quad (34)$$

First suppose $P \not\ll Q$. Then there exists a measurable set $A$ with $Q(A) = 0$ and $P(A) > 0$. Let $g_n = -n1_A - 1_{A^c}$. Then

$$\sup_{g \in \mathcal{M}_b(\Omega):g<0} \left\{ \int g dQ + \log \int |g| dP \right\} + 1 \geq \int g_n dQ + \log \int |g_n| dP + 1 \quad (35)$$

$$= -nQ(A) - Q(A^c) + \log(nP(A) + P(A^c)) + 1 = \log(nP(A) + P(A^c)) \to \infty \quad (36)$$

as $n \to \infty$. Therefore

$$\sup_{g \in \mathcal{M}_b(\Omega):g<0} \left\{ \int g dQ + \log \int |g| dP \right\} + 1 = \infty = D_\infty(P\|Q). \quad (37)$$

Now suppose $P \ll Q$. Using the definition (32) along with Theorem A.2 and changing variables $g = \tilde{g}/\alpha$ we have

$$D_\infty(P\|Q) = \lim_{\alpha\to\infty} \alpha R_\alpha(P\|Q) \quad (38)$$

$$= \lim_{\alpha\to\infty} \left[ \sup_{g \in \mathcal{M}_b(\Omega):g<0} \left\{ \int \alpha g dQ + \frac{\alpha}{\alpha-1} \log \int |g|^{(\alpha-1)/\alpha} dP \right\} + (\log\alpha + 1) \right]$$

$$\geq \lim_{\alpha\to\infty} \left[ \int \tilde{g} dQ + \frac{\alpha}{\alpha-1} \log \int |\tilde{g}/\alpha|^{(\alpha-1)/\alpha} dP + (\log\alpha + 1) \right]$$

$$= \lim_{\alpha\to\infty} \left[ \int \tilde{g} dQ + \frac{\alpha}{\alpha-1} \log \int |\tilde{g}|^{(\alpha-1)/\alpha} dP + 1 \right]$$

$$= \int \tilde{g} dQ + \log \int |\tilde{g}| dP + 1 \text{ for all } \tilde{g} \in \mathcal{M}_b(\Omega), \tilde{g} < 0.$$

Here we used the dominated convergence theorem to evaluate the limit. Hence, by maximizing over $\tilde{g}$ we obtain

$$D_\infty(P\|Q) \geq \sup_{g \in \mathcal{M}_b(\Omega):g<0} \left\{ \int g dQ + \log \int |g| dP \right\} + 1. \quad (39)$$

To prove the reverse inequality, take any $r \in (0, \text{ess sup}_P dP/dQ)$. By definition of the essential supremum we have $P(dP/dQ > r) > 0$. We also have the bound

$$P(dP/dQ > r) = \int 1_{dP/dQ > r} \frac{dP}{dQ} dQ \geq \int 1_{dP/dQ > r} r dQ = rQ(dP/dQ > r). \tag{40}$$

For $c, \epsilon > 0$ define $g_{c,\epsilon} = -c1_{dP/dQ > r} - \epsilon$. These satisfy $g_{c,\epsilon} \in \mathcal{M}_b(\Omega)$, $g_{c,\epsilon} < 0$ and so

$$\sup_{g \in \mathcal{M}_b(\Omega):g<0} \left\{ \int g dQ + \log \int |g| dP \right\} + 1 \geq \int g_{c,\epsilon} dQ + \log \int |g_{c,\epsilon}| dP + 1 \tag{41}$$

$$= -cQ(dP/dQ > r) - \epsilon + \log(cP(dP/dQ > r) + \epsilon) + 1$$

$$\geq -cP(dP/dQ > r)/r - \epsilon + \log(cP(dP/dQ > r) + \epsilon) + 1.$$

Letting $\epsilon \to 0^+$ we find

$$\sup_{g \in \mathcal{M}_b(\Omega):g<0} \left\{ \int g dQ + \log \int |g| dP \right\} + 1 \geq -cP(dP/dQ > r)/r + \log(cP(dP/dQ > r)) + 1 \tag{42}$$

for all $c > 0$. We have $P(dP/dQ > r) > 0$, hence by maximizing over $c > 0$ and changing variables to $z = cP(dP/dQ > r)$ we obtain

$$\sup_{g \in \mathcal{M}_b(\Omega):g<0} \left\{ \int g dQ + \log \int |g| dP \right\} + 1 \geq \sup_{z>0} \{-z/r + \log(z) + 1\} = \log(r). \tag{43}$$

This holds for all $r < \text{ess sup}_P dP/dQ$, therefore we can take $r \nearrow \text{ess sup}_P dP/dQ$ and use (34) to conclude

$$\sup_{g \in \mathcal{M}_b(\Omega):g<0} \left\{ \int g dQ + \log \int |g| dP \right\} + 1 \geq \log(\text{ess sup}_P dP/dQ) = D_\infty(P\|Q). \tag{44}$$

Combining this with (39) completes the proof of (33).

Now suppose $\Omega$ is a metric space. We clearly have

$$D_\infty(P\|Q) = \sup_{g \in \mathcal{M}_b(\Omega):g<0} \left\{ \int g dQ + \log \int |g| dP \right\} + 1 \tag{45}$$

$$\geq \sup_{g \in C_b(\Omega):g<0} \left\{ \int g dQ + \log \int |g| dP \right\} + 1.$$

To prove the reverse inequality, take any $g \in \mathcal{M}_b(\Omega)$ with $g < 0$. By Lusin's theorem, for all $\epsilon > 0$ there exists a closed set $E_\epsilon$ and $h_\epsilon \in C_b(\Omega)$ such that $P(E_\epsilon^c) \leq \epsilon$, $Q(E_\epsilon^c) \leq \epsilon$, $h_\epsilon|_{E_\epsilon} = g$, and $\inf g \leq h_\epsilon \leq 0$. Define $g_\epsilon = h_\epsilon - \epsilon$. Then $g_\epsilon < 0$, $g_\epsilon \in C_b(\Omega)$ and we have

$$\sup_{g \in C_b(\Omega):g<0} \left\{ \int g dQ + \log \int |g| dP \right\} \geq \int g_\epsilon dQ + \log \int |g_\epsilon| dP \tag{46}$$

$$= \int g dQ + \int (h_\epsilon - g) 1_{E_\epsilon^c} dQ - \epsilon + \log(\int |g| dP + \int (|h_\epsilon| - |g|) 1_{E_\epsilon^c} dP + \epsilon)$$

$$\geq \int g dQ - (\sup g - \inf g) Q(E_\epsilon^c) - \epsilon + \log(\int |g| dP + \inf g P(E_\epsilon^c) + \epsilon)$$

$$\geq \int g dQ - (\sup g - \inf g)\epsilon - \epsilon + \log(\int |g| dP + \inf g \epsilon + \epsilon).$$

Taking the limit $\epsilon \to 0^+$ we therefore obtain

$$\sup_{g \in C_b(\Omega):g<0} \left\{ \int g dQ + \log \int |g| dP \right\} \geq \int g dQ + \log \int |g| dP. \tag{47}$$

This holds for all $g \in \mathcal{M}_b(\Omega)$ with $g < 0$, hence by taking the supremum over $g$ we obtain the reverse inequality to (45). This completes the proof. $\square$

## B  PROOFS

In this appendix we provide a number of proofs that were omitted from the main text. Recall that $X$ denotes a compact metric space.

**Lemma B.1.** *Let $\Gamma \subset C(X)$ and $P, Q \in \mathcal{P}(X)$. Then $\alpha R_\alpha^{\Gamma/\alpha, IC}(P\|Q)$ is non-decreasing in $\alpha \in (0, 1) \cup (1, \infty)$. If $0 \in \Gamma$ then $\alpha R_\alpha^{\Gamma, IC}(P\|Q)$ is also non-decreasing.*

*Proof.* If $0 \in \Gamma$ then $W^\Gamma \geq 0$, hence

$$\alpha R_\alpha^{\Gamma, IC}(P\|Q) = \inf_{\eta \in \mathcal{P}(X)} \{\alpha R_\alpha(P\|\eta) + \alpha W^\Gamma(Q, \eta)\} \tag{48}$$

where both $\alpha \mapsto \alpha R_\alpha(P\|\eta)$ and $\alpha \mapsto \alpha W^\Gamma(Q, \eta)$ are non-decreasing. Therefore the infimum is as well. The proof for $\alpha \mapsto \alpha R_\alpha^{\Gamma/\alpha, IC}(P\|Q)$ is similar, though it doesn't require the assumption $0 \in \Gamma$ due to the identity $\alpha W^{\Gamma/\alpha} = W^\Gamma$. $\qquad\square$

Next we prove a key lemma that is used in our main result. First recall the definition

$$\Lambda_\alpha^P[g] := \infty 1_{g \not< 0} - \left(\frac{1}{\alpha - 1} \log \int |g|^{(\alpha-1)/\alpha} dP + \alpha^{-1}(\log \alpha + 1)\right) 1_{g < 0}, \quad g \in C(X). \tag{49}$$

**Lemma B.2.** *$\Lambda_\alpha^P$ is convex and is continuous on $\{g \in C(X) : g < 0\}$, an open subset of $C(X)$.*

*Proof.* First we prove convexity. Let $g_0, g_1 \in \{C(X) : g < 0\}$ and $\lambda \in (0, 1)$. For $\alpha \in (0, 1)$ we can use the inequality $\lambda a + (1 - \lambda)b \geq a^\lambda b^{1-\lambda}$ for all $a, b > 0$ to compute

$$-\frac{1}{\alpha - 1} \log \int |\lambda g_1 + (1 - \lambda)g_0|^{(\alpha-1)/\alpha} dP \leq -\frac{1}{\alpha - 1} \log \int (|g_1|^\lambda |g_0|^{1-\lambda})^{(\alpha-1)/\alpha} dP. \tag{50}$$

Using Hölder's inequality with exponents $p = 1/\lambda$, $q = 1/(1 - \lambda)$ we then obtain

$$-\frac{1}{\alpha - 1} \log \int (|g_1|^\lambda |g_0|^{1-\lambda})^{(\alpha-1)/\alpha} dP \tag{51}$$

$$\leq -\frac{1}{\alpha - 1} \log \left(\int |g_1|^{(\alpha-1)/\alpha} dP^\lambda \int |g_0|^{(\alpha-1)/\alpha} dP^{1-\lambda}\right)$$

$$= \lambda \left(-\frac{1}{\alpha - 1} \log \int |g_1|^{(\alpha-1)/\alpha} dP\right) + (1 - \lambda) \left(-\frac{1}{\alpha - 1} \log \int |g_0|^{(\alpha-1)/\alpha} dP\right).$$

Therefore $g \mapsto -\frac{1}{\alpha-1} \log \int |g|^{(\alpha-1)/\alpha} dP$ is convex on $\{g < 0\}$. This proves $\Lambda_\alpha^P$ is convex when $\alpha \in (0, 1)$.

Now suppose $\alpha > 1$. The map $t > 0, t \mapsto t^{(\alpha-1)/\alpha}$ is concave and $-\log$ is decreasing and convex, hence

$$-\frac{1}{\alpha - 1} \log \int |\lambda g_1 + (1 - \lambda)g_0|^{(\alpha-1)/\alpha} dP \tag{52}$$

$$\leq -\frac{1}{\alpha - 1} \log \left(\lambda \int |g_1|^{(\alpha-1)/\alpha} dP + (1 - \lambda) \int |g_0|^{(\alpha-1)/\alpha} dP\right)$$

$$\leq \lambda \left(-\frac{1}{\alpha - 1} \log \int |g_1|^{(\alpha-1)/\alpha} dP\right) + (1 - \lambda) \left(-\frac{1}{\alpha - 1} \log \int |g_0|^{(\alpha-1)/\alpha} dP\right).$$

This proves that $\Lambda_\alpha^P$ is also convex when $\alpha > 1$. Openness of $\{g < 0\}$ follows from the assumption that $X$ is compact and so any strictly negative continuous function is strictly bounded away from zero. Continuity on $\{g < 0\}$ then follows from the dominated convergence theorem. $\qquad\square$

Now we prove our main theorem, deriving the dual variational formula and other important properties of the IC-$\Gamma$-Rényi divergences.

**Theorem B.3.** *Let $\Gamma \subset C(X)$ be admissible, $P, Q \in \mathcal{P}(X)$, and $\alpha \in (0, 1) \cup (1, \infty)$. Then:*

*1.*

$$R_\alpha^{\Gamma,IC}(P\|Q) = \sup_{g\in\Gamma:g<0}\left\{\int gdQ + \frac{1}{\alpha-1}\log\int|g|^{(\alpha-1)/\alpha}dP\right\} + \alpha^{-1}(\log\alpha+1)\,. \tag{53}$$

2. *If (53) is finite then there exists $\eta_*\in\mathcal{P}(X)$ such that*

$$R_\alpha^{\Gamma,IC}(P\|Q) = \inf_{\eta\in\mathcal{P}(X)}\{R_\alpha(P\|\eta) + W^\Gamma(Q,\eta)\} = R_\alpha(P\|\eta_*) + W^\Gamma(Q,\eta_*)\,. \tag{54}$$

3. $R_\alpha^{\Gamma,IC}(P\|Q)$ *is convex in $Q$. If $\alpha\in(0,1)$ then $R_\alpha^{\Gamma,IC}(P\|Q)$ is jointly convex in $(P,Q)$.*

4. $(P,Q)\mapsto R_\alpha^{\Gamma,IC}(P\|Q)$ *is lower semicontinuous.*

5. $R_\alpha^{\Gamma,IC}(P\|Q)\geq 0$ *with equality if $P=Q$.*

6. $R_\alpha^{\Gamma,IC}(P\|Q)\leq \min\{R_\alpha(P\|Q), W^\Gamma(Q,P)\}.$

7. *If $\Gamma$ is strictly admissible then $R_\alpha^{\Gamma,IC}$ has the divergence property.*

*Proof.*        1. Define $F, G : C(X)\to(-\infty,\infty]$ by $F = \Lambda_\alpha^P$ and $G[g] = \infty 1_{g\notin\Gamma} - E_Q[g]$. Using the assumptions on $\Gamma$ along with Lemma B.2 we see that $F$ and $G$ are convex, $F[-1]<\infty$, $G[-1]<\infty$, and $F$ is continuous at $-1$. Therefore Fenchel-Rockafellar duality (see, e.g., Theorem 4.4.3 in Borwein & Zhu (2006)) along with the identity $C(X)^* = M(X)$ gives

$$\sup_{g\in C(X)}\{-F[g] - G[g]\} = \inf_{\eta\in M(X)}\{F^*[\eta] + G^*[-\eta]\}\,, \tag{55}$$

and if either side is finite then the infimum on the right hand side is achieved at some $\eta_*\in M(X)$. Using the definitions, we can rewrite the left hand side as follow

$$\sup_{g\in C(X)}\{-F[g] - G[g]\} \tag{56}$$
$$= \sup_{g\in\Gamma:g<0}\left\{\int gdQ + \frac{1}{\alpha-1}\log\int|g|^{(\alpha-1)/\alpha}dP\right\} + \alpha^{-1}(\log\alpha+1)\,.$$

We can also compute

$$G^*[-\eta] = \sup_{g\in C(X)}\left\{-\int gd\eta - (\infty 1_{g\notin\Gamma} - E_Q[g])\right\} = W^\Gamma(Q,\eta)\,. \tag{57}$$

Therefore

$$\inf_{\eta\in M(X)}\{(\Lambda_\alpha^P)^*[\eta] + W^\Gamma(Q,\eta)\} \tag{58}$$
$$= \sup_{g\in\Gamma:g<0}\left\{\int gdQ + \frac{1}{\alpha-1}\log\int|g|^{(\alpha-1)/\alpha}dP\right\} + \alpha^{-1}(\log\alpha+1)\,.$$

Next we show that the infimum over $M(X)$ can be restricted to $\mathcal{P}(X)$. First suppose $\eta\in M(X)$ with $\eta(X)\neq 1$. Then, using the assumption that $\Gamma$ contains the constant functions, we have

$$W^\Gamma(Q,\eta)\geq E_Q[\pm n] - \int\pm nd\eta = \pm n(1 - \eta(X))\to\infty \tag{59}$$

as $n\to\infty$ (for appropriate choice of sign). Therefore $W^\Gamma(Q,\eta) = \infty$ if $\eta(X)\neq 1$. This implies that the infimum can be restricted to $\{\eta\in M(X):\eta(X) = 1\}$.

Now suppose $\eta\in M(X)$ is not positive. Take a measurable set $A$ with $\eta(A)<0$. By Lusin's theorem, for all $\epsilon>0$ there exists a closed set $E_\epsilon\subset X$ and a continuous function

$g_\epsilon \in C(X)$ such that $|\eta|(E_\epsilon^c) < \epsilon$, $0 \le g_\epsilon \le 1$, and $g_\epsilon|_{E_\epsilon} = 1_A$. Define $g_{n,\epsilon} = -ng_\epsilon - 1$, $n \in \mathbb{Z}^+$. Then $g_{n,\epsilon} \in \{g \in C(X) : g < 0\}$, hence

$$(\Lambda_\alpha^P)^*[\eta] \ge \int g_{n,\epsilon} d\eta + \frac{1}{\alpha - 1} \log \int |g_{n,\epsilon}|^{(\alpha-1)/\alpha} dP + \alpha^{-1}(\log \alpha + 1) \tag{60}$$

$$= -n\eta(A) + n\eta(A \cap E_\epsilon^c) - n \int g_\epsilon 1_{E_\epsilon^c} d\eta - \eta(X)$$

$$+ \frac{1}{\alpha - 1} \log \int |ng_\epsilon + 1|^{(\alpha-1)/\alpha} dP + \alpha^{-1}(\log \alpha + 1)$$

$$\ge n(|\eta(A)| - 2\epsilon) - \eta(X) + \frac{1}{\alpha - 1} \log \int |ng_\epsilon + 1|^{(\alpha-1)/\alpha} dP + \alpha^{-1}(\log \alpha + 1).$$

If $\alpha > 1$ then $\log \int |ng_\epsilon + 1|^{(\alpha-1)/\alpha} dP \ge 0$ and if $\alpha \in (0,1)$ then $\log \int |ng_\epsilon + 1|^{(\alpha-1)/\alpha} dP \le 0$. In either case we have $\frac{1}{\alpha-1} \log \int |ng_\epsilon + 1|^{(\alpha-1)/\alpha} dP \ge 0$ and so

$$(\Lambda_\alpha^P)^*[\eta] \ge n(|\eta(A)| - 2\epsilon) - \eta(X) + \alpha^{-1}(\log \alpha + 1). \tag{61}$$

By choosing $\epsilon < |\eta(A)|/2$ and taking $n \to \infty$ we see that $(\Lambda_\alpha^P)^*[\eta] = \infty$ whenever $\eta \in M(X)$ is not positive. Therefore the infimum can further be restricted to positive measures. Combining these results we find

$$\sup_{g \in \Gamma : g < 0} \left\{ \int g dQ + \frac{1}{\alpha - 1} \log \int |g|^{(\alpha-1)/\alpha} dP \right\} + \alpha^{-1}(\log \alpha + 1) \tag{62}$$

$$= \inf_{\eta \in \mathcal{P}(X)} \{(\Lambda_\alpha^P)^*[\eta] + W^\Gamma(Q, \eta)\}.$$

For $\eta \in \mathcal{P}(X)$, equation (10) implies $(\Lambda_\alpha^P)^*[\eta] = R_\alpha(P\|\eta)$. This completes the proof.

2. The existence of a minimizer follows from Fenchel-Rockafellar duality; again, see Theorem 4.4.3 in Borwein & Zhu (2006).

3. This follows from (53) together with the fact that the supremum of convex functions is convex and $y \mapsto \frac{1}{\alpha-1} \log(y)$ is convex when $\alpha \in (0,1)$.

4. Compactness of $X$ implies that $g$ and $|g|^{(\alpha-1)/\alpha}$ are bounded and continuous whenever $g \in \Gamma$ satisfies $g < 0$. Therefore $Q \to \int g dQ$ and $P \to \int |g|^{(\alpha-1)/\alpha} dP$ are continuous in the weak topology on $\mathcal{P}(X)$. Therefore the objective functional in (53) is continuous in $(P, Q)$. The supremum is therefore lower semicontinuous.

5. This easily follows from the definition (7).

6. $R_\alpha$ is a divergence, hence is non-negative. $\Gamma$ contains the constant functions, hence $W^\Gamma \ge 0$. Therefore $R_\alpha^{\Gamma, IC} \ge 0$. If $Q = P$ then $0 \le R_\alpha^{\Gamma, IC}(P\|Q) \le R_\alpha(P\|P) + W^\Gamma(P, P) = 0$, hence $R_\alpha^{\Gamma, IC}(P\|Q) = 0$.

7. Suppose $\Gamma$ is strictly admissible. Due to part 5 of this theorem, we only need to show that if $R_\alpha^{\Gamma, IC}(P\|Q) = 0$ then $P = Q$. If $R_\alpha^{\Gamma, IC}(P\|Q) = 0$ then part 2 implies there exists $\eta_* \in \mathcal{P}(X)$ such that

$$0 = R_\alpha(P\|\eta_*) + W^\Gamma(Q, \eta_*). \tag{63}$$

Both terms are non-negative, hence $R_\alpha(P\|\eta_*) = 0 = W^\Gamma(Q, \eta_*)$. $R_\alpha$ has the divergence property, hence $\eta_* = P$. So $W^\Gamma(Q, P) = 0$. Therefore $0 \ge \int g dQ - \int g dP$ for all $g \in \Gamma$. Let $\Psi$ be as in the definition of strict admissibility and let $\psi \in \Psi$. There exists $c \in \mathbb{R}$, $\epsilon > 0$ such that $c \pm \epsilon\psi \in \Gamma$ and so $0 \ge \pm\epsilon(\int \psi dQ - \int \psi dP)$. Therefore $\int \psi dQ = \int \psi dP$ for all $\psi \in \Psi$. $\Psi$ is $\mathcal{P}(X)$-determining, hence $Q = P$.

$\square$

**Remark B.4.** *The Fenchel-Rockafellar Theorem applies under two different sets of assumptions: the first assumes both mappings are lower semicontinuous (LSC) while the second applies when one mapping is continuous at a point where both are finite. The mapping $\Lambda_\alpha^P$, as defined by (49) and*

*appearing in (10), is **not** LSC but it is continuous on its domain, hence we used the second version of Fenchel-Rockafellar in our proof of Theorem B.3. For $\alpha > 1$ one could alternatively redefine $\Lambda_\alpha^P$ along the boundary of $\{g < 0\}$ to make it LSC while still maintaining the relation (10) and thereby utilize the first version of Fenchel-Rockafellar. This alternative approach is also amenable to extending the theorem to non-compact spaces, using the methods from Dupuis, Paul & Mao, Yixiang (2022); Birrell et al. (2022a). However, these methods do not apply to $\alpha \in (0, 1)$. With this in mind, in order to provide a simple unified treatment of all $\alpha \in (0, 1) \cup (1, \infty)$ we structured our proof around the second version of Fenchel-Rockafellar.*

*Despite the fact that $\Lambda_\alpha^P$ is not LSC, the Fenchel-Rockafellar Theorem does imply that convex duality holds at all points of continuity in the domain, i.e., one has*

$$\Lambda_\alpha^P[g] = \sup_{\eta \in M(X)} \{ \int g d\eta - R_\alpha(P\|\eta)\} \ \text{for all } g < 0, \tag{64}$$

*but this duality formula doesn't necessarily hold if $g \not< 0$. Here, $R_\alpha(P\|\eta)$ for general $\eta \in M(X)$ is defined via the variational formula*

$$R_\alpha(P\|\eta) := (\Lambda_\alpha^P)^*[\eta] = \sup_{g \in C(X)} \{ \int g d\eta - \Lambda_\alpha^P[g]\} \tag{65}$$

*and one can rewrite this in terms of the classical Rényi divergence as follows*

$$R_\alpha(P\|\eta) = \begin{cases} \infty \ \text{if } \eta \not\geq 0 \ \text{or } \eta = 0, \\ R_\alpha(P\|\frac{\eta}{\|\eta\|}) - \frac{1}{\alpha}\log\|\eta\| \ \text{if } \eta \text{ is a nonzero positive measure.} \end{cases} \tag{66}$$

Next we prove the limiting results from Theorem 4.1.

**Theorem B.5.** *Let $\Gamma \subset C(X)$ be admissible, $P, Q \in \mathcal{P}(X)$, and $\alpha \in (0, 1) \cup (1, \infty)$. Then*

$$\lim_{\delta \to 0^+} \frac{1}{\delta} R_\alpha^{\delta\Gamma, IC}(P\|Q) = W^\Gamma(Q, P) \tag{67}$$

*and if $\Gamma$ is strictly admissible we have*

$$\lim_{L \to \infty} R_\alpha^{L\Gamma, IC}(P\|Q) = R_\alpha(P\|Q). \tag{68}$$

*Proof.* It is straightforward to show that the scaled function spaces are admissible and $W^{c\Gamma} = cW^\Gamma$ for all $c > 0$. First we prove (67). From the definition 7 we have

$$\delta^{-1} R_\alpha^{\delta\Gamma, IC}(P\|Q) = \inf_{\eta \in \mathcal{P}(X)} \{\delta^{-1} R_\alpha(P\|\eta) + W^\Gamma(Q, \eta)\} \leq W^\Gamma(Q, P) \tag{69}$$

and so $\delta^{-1} R_\alpha^{\delta\Gamma, IC}(P\|Q)$ is non-increasing in $\delta$. Therefore

$$\lim_{\delta \to 0^+} \delta^{-1} R_\alpha^{\delta\Gamma, IC}(P\|Q) = \sup_{\delta > 0} \delta^{-1} R_\alpha^{\delta\Gamma, IC}(P\|Q) \tag{70}$$

and

$$\lim_{\delta \to 0^+} \delta^{-1} R_\alpha^{\delta\Gamma, IC}(P\|Q) \leq W^\Gamma(Q, P). \tag{71}$$

We will assume this inequality is strict and derive a contradiction. This assumption, together with (70), implies $R_\alpha^{\delta\Gamma, IC}(P\|Q) < \infty$ for all $\delta > 0$. Part (2) of Theorem 3.4 then implies the existence of $\eta_{*,\delta} \in \mathcal{P}(X)$ such that

$$\delta^{-1} R_\alpha^{\delta\Gamma, IC}(P\|Q) = \delta^{-1} R_\alpha(P\|\eta_{*,\delta}) + W^\Gamma(Q, \eta_{*,\delta}) \geq W^\Gamma(Q, \eta_{*,\delta}). \tag{72}$$

Take a sequence $\delta_n \to 0^+$. We have assumed $X$ is compact, hence $\mathcal{P}(X)$ is also compact and so there exists a weakly convergent subsequence $\eta_{*,\delta_{n_j}} \to \eta_*$. From the variational formulas (25) and (8) we see that $R_\alpha(P\|\cdot)$ and $W^\Gamma(Q, \cdot)$ are lower semicontinuous, hence $\liminf_j W^\Gamma(Q, \eta_{*,\delta_{n_j}}) \geq W^\Gamma(Q, \eta_*)$ and

$$R_\alpha(P\|\eta_*) \leq \liminf_j R_\alpha(P\|\eta_{*,\delta_{n_j}}) \leq \liminf_j \delta_{n_j}(\delta_{n_j}^{-1} R_\alpha(P\|\eta_{*,\delta_{n_j}}) + W^\Gamma(Q, \eta_{*,\delta_{n_j}})) \tag{73}$$

$$= \liminf_j \delta_{n_j}(\delta_{n_j}^{-1} R_\alpha^{\delta_{n_j}\Gamma, IC}(P\|Q)) = 0, \tag{74}$$

where the last equality follows from the assumed strictness of the inequality (70). Therefore the divergence property for the classical Rényi divergences implies $R_\alpha(P\|\eta_*) = 0$ and $P = \eta_*$. Combining the above results we obtain

$$\lim_{\delta\to 0^+} \delta^{-1} R_\alpha^{\delta\Gamma,IC}(P\|Q) = \lim_{j\to\infty} \delta_{n_j}^{-1} R_\alpha^{\delta_{n_j}\Gamma,IC}(P\|Q) \geq \liminf_j W^\Gamma(Q, \eta_{*,\delta_{n_j}}) \tag{75}$$

$$\geq W^\Gamma(Q, \eta_*) = W^\Gamma(Q, P).$$

This contradicts (71) and therefore we have proven the equality (67).

Now we assume $\Gamma$ is strictly admissible and will prove (68) via similar reasoning. From the definition 7 we see that

$$R_\alpha^{L\Gamma,IC}(P\|Q) = \inf_{\eta\in\mathcal{P}(X)}\{R_\alpha(P\|\eta) + LW^\Gamma(Q,\eta)\} \leq R_\alpha(P\|Q) \tag{76}$$

and $R_\alpha^{L\Gamma,IC}$ is non-decreasing in $L$. Hence $\lim_{L\to\infty} R_\alpha^{L\Gamma,IC}(P\|Q) = \sup_{L>0} R_\alpha^{L\Gamma,IC}(P\|Q)$ and

$$\lim_{L\to\infty} R_\alpha^{L\Gamma,IC}(P\|Q) \leq R_\alpha(P\|Q). \tag{77}$$

Suppose this inequality is strict. Then $R_\alpha^{L\Gamma,IC}(P\|Q) < \infty$ for all $L$ and we can use part (2) of Theorem 3.4 to conclude there exists $\eta_{*,L} \in \mathcal{P}(X)$ such that

$$R_\alpha^{L\Gamma,IC}(P\|Q) = R_\alpha(P\|\eta_{*,L}) + LW^\Gamma(Q,\eta_{*,L}). \tag{78}$$

Take $L_n \to \infty$. Compactness of $\mathcal{P}(X)$ implies the existence of a weakly convergent subsequence $\eta_{*,j} := \eta_{*,L_{n_j}} \to \eta_* \in \mathcal{P}(X)$. Lower semicontinuity of $R_\alpha(P\|\cdot)$ and $W^\Gamma(Q,\cdot)$ imply $\liminf_j R_\alpha(P\|\eta_{*,j}) \geq R_\alpha(P\|\eta_*)$ and

$$W^\Gamma(Q,\eta_*) \leq \liminf_j W^\Gamma(Q,\eta_{*,j}) = \liminf_j L_{n_j}^{-1} W^{L_{n_j}\Gamma}(Q,\eta_{*,j}) \tag{79}$$

$$\leq \liminf_j L_{n_j}^{-1} R_\alpha^{L_{n_j}\Gamma,IC}(P\|Q) = 0,$$

where the last equality follows from the assumed strictness of the inequality (77). Therefore $W^\Gamma(Q,\eta_*) = 0$. $\Gamma$ is strictly admissible, hence $Q = \eta_*$ (see the proof of part (7) of Theorem 3.4). Combining these results we see that

$$\lim_{L\to\infty} R_\alpha^{L\Gamma,IC}(P\|Q) = \lim_j R_\alpha^{L_{n_j}\Gamma,IC}(P\|Q) = \lim_j (R_\alpha(P\|\eta_{*,L_{n_j}}) + L_{n_j} W^\Gamma(Q,\eta_{*,L_{n_j}})) \tag{80}$$

$$\geq \liminf_j R_\alpha(P\|\eta_{*,j}) \geq R_\alpha(P\|\eta_*) = R_\alpha(P\|Q).$$

This contradicts the assumed strictness of the inequality (77) and hence (77) is an equality. This completes the proof.

$\square$

Next we prove Theorem 4.2, regarding the $\alpha \to 1$ limit of the IC-$\Gamma$-Rényi divergences.

**Theorem B.6.** *Let $\Gamma \subset C(X)$ be admissible and $P, Q \in \mathcal{P}(X)$. Then*

$$\lim_{\alpha\to 1^+} R_\alpha^{\Gamma,IC}(P\|Q) = \inf_{\substack{\eta\in\mathcal{P}(X):\\ \exists\beta>1, R_\beta(P\|\eta)<\infty}} \{R(P\|\eta) + W^\Gamma(Q,\eta)\}, \tag{81}$$

$$\lim_{\alpha\to 1^-} R_\alpha^{\Gamma,IC}(P\|Q) = \inf_{\eta\in\mathcal{P}(X)} \{R(P\|\eta) + W^\Gamma(Q,\eta)\} \tag{82}$$

$$= \sup_{g\in\Gamma:g<0} \{\int g\,dQ + \int \log|g|\,dP\} + 1. \tag{83}$$

*Proof.* Lemma B.1 implies $\alpha \mapsto \alpha R_\alpha^{\Gamma,IC}(P\|Q)$ is non-decreasing on $(1,\infty)$, therefore

$$\lim_{\alpha\to 1^+} \alpha R_\alpha^{\Gamma,IC}(P\|Q) = \inf_{\alpha>1} \alpha R_\alpha^{\Gamma,IC}(P\|Q) \tag{84}$$

$$= \inf_{\alpha>1} \inf_{\eta\in\mathcal{P}(X)} \{\alpha R_\alpha(P\|\eta) + \alpha W^\Gamma(Q,\eta)\}$$

$$= \inf_{\eta\in\mathcal{P}(X)} \inf_{\alpha>1} \{\alpha R_\alpha(P\|\eta) + \alpha W^\Gamma(Q,\eta)\}$$

$$= \inf_{\eta\in\mathcal{P}(X)} \{\lim_{\alpha\to 1^+} \alpha R_\alpha(P\|\eta) + W^\Gamma(Q,\eta)\}.$$

From Van Erven & Harremos (2014) we have

$$\lim_{\alpha \to 1^+} R_\alpha(P\|\eta) = \begin{cases} R(P\|\eta) & \text{if } \exists \beta > 1, R_\beta(P\|\eta) < \infty \\ \infty & \text{otherwise} \end{cases} \tag{85}$$

and so we can conclude

$$\lim_{\alpha \to 1^+} R_\alpha^{\Gamma, IC}(P\|Q) = \lim_{\alpha \to 1^+} \alpha R_\alpha^{\Gamma, IC}(P\|Q) = \inf_{\substack{\eta \in P(X): \\ \exists \beta > 1, R_\beta(P\|\eta) < \infty}} \{R(P\|\eta) + W^\Gamma(Q, \eta)\}. \tag{86}$$

This proves (81).

Now we compute the limit as $\alpha \to 1^-$. Note that the limit exists due to the fact that $\alpha \mapsto \alpha R_\alpha^{\Gamma, IC}(P\|Q)$ is non-decreasing. From the definition (7), for all $\eta \in \mathcal{P}(X)$ we have

$$\lim_{\alpha \to 1^-} R_\alpha^{\Gamma, IC}(P\|Q) \le \lim_{\alpha \to 1^-} R_\alpha(P\|\eta) + W^\Gamma(Q, \eta) = R(P\|\eta) + W^\Gamma(Q, \eta). \tag{87}$$

Here we used the fact that $\lim_{\alpha \to 1^-} R_\alpha(P\|\eta) = R(P\|\eta)$ (see Van Erven & Harremos (2014)). Maximizing over $\eta$ then gives

$$\lim_{\alpha \to 1^-} R_\alpha^{\Gamma, IC}(P\|Q) \le \inf_{\eta \in \mathcal{P}(X)} \{R(P\|\eta) + W^\Gamma(Q, \eta)\}. \tag{88}$$

To prove the reverse inequality, use part 1 of Theorem 3.4 to compute

$$\lim_{\alpha \to 1^-} R_\alpha^{\Gamma, IC}(P\|Q) = \lim_{\alpha \to 1^-} \alpha R_\alpha^{\Gamma, IC}(P\|Q) \tag{89}$$

$$= \lim_{\alpha \to 1^-} \sup_{g \in \Gamma: g < 0} \left\{ \alpha \int g dQ + \frac{\alpha}{\alpha - 1} \log \int |g|^{(\alpha-1)/\alpha} dP + \log \alpha + 1 \right\}$$

$$\ge \int g dQ + \lim_{\alpha \to 1^-} \frac{\alpha}{\alpha - 1} \log \int |g|^{(\alpha-1)/\alpha} dP + 1$$

$$= \int g dQ + \frac{d}{dy}\Big|_{y=0} \log \int e^{y \log |g|} dP + 1$$

$$= \int g dQ + \int \log |g| dP + 1$$

for all $g \in \Gamma$, $g < 0$. Therefore, maximizing over $g$ gives

$$\lim_{\alpha \to 1^-} R_\alpha^{\Gamma, IC}(P\|Q) \ge \sup_{g \in \Gamma: g < 0} \left\{ \int g dQ + \int \log |g| dP \right\} + 1. \tag{90}$$

We now use Fenchel-Rockafellar duality (Theorem 4.4.3 in Borwein & Zhu (2006)) to compute the dual variational representation of the right hand side of (90). Define $F, G : C(X) \to (-\infty, \infty]$ by $F[g] = \infty 1_{g \not< 0} - \int \log |g| dP$ and $G[g] = \infty 1_{g \notin \Gamma} - E_Q[g]$. It is straightforward to show that $F$ and $G$ are convex, $F[-1] < \infty$, $G[-1] < \infty$, and $F$ is continuous at $-1$. Therefore

$$\inf_{g \in C(X)} \{F[g] + G[g]\} = \sup_{\eta \in E^*} \{-F^*(-\eta) - G^*(\eta)\}, \tag{91}$$

i.e.

$$\sup_{g \in \Gamma: g < 0} \left\{ E_Q[g] + \int \log |g| dP \right\} = \inf_{\eta \in M(X)} \{F^*(\eta) + W^\Gamma(Q, \eta)\}, \tag{92}$$

where $F^*(\eta) = \sup_{g \in C(X): g < 0} \{\int g d\eta + \int \log |g| dP\}$. Now we show the infimum can be restricted to $\eta \in \mathcal{P}(X)$: If $\eta(X) \ne 1$ then by taking $g = \pm n$ we find

$$W^\Gamma(Q, \eta) \ge n|Q(X) - \eta(X)| \to \infty \tag{93}$$

as $n \to \infty$. Therefore $W^\Gamma(Q, \eta) = \infty$ if $\eta(X) \ne 1$.

Now suppose $\eta \in M(X)$ is not positive. Take a measurable set $A$ with $\eta(A) < 0$. By Lusin's theorem, for all $\epsilon > 0$ there exists a closed set $E_\epsilon \subset X$ and a continuous function $g_\epsilon \in C(X)$

such that $|\eta|(E_\epsilon^c) < \epsilon$, $0 \le g_\epsilon \le 1$, and $g_\epsilon|_{E_\epsilon} = 1_A$. Define $g_{n,\epsilon} = -ng_\epsilon - 1$, $n \in \mathbb{Z}^+$. Then $g_{n,\epsilon} \in \{g \in C(X) : g < 0\}$, hence

$$F^*(\eta) \ge \int g_{n,\epsilon} d\eta + \int \log|g_{n,\epsilon}| dP \tag{94}$$

$$= \int -ng_\epsilon - 1 d\eta + \int \log(ng_\epsilon + 1) dP$$

$$\ge n(|\eta(A)| - \int (g_\epsilon - 1_A) 1_{E_\epsilon^c} d\eta) - \eta(X)$$

$$\ge n(|\eta(A)| - \epsilon) - \eta(X).$$

Letting $\epsilon = |\eta(A)|/2$ and taking $n \to \infty$ gives $F^*(\eta) = \infty$. Therefore we conclude

$$\inf_{\eta \in M(X)} \{F^*(\eta) + W^\Gamma(Q, \eta)\} = \inf_{\eta \in \mathcal{P}(X)} \{F^*(\eta) + W^\Gamma(Q, \eta)\}. \tag{95}$$

To evaluate $F^*(\eta)$ for $\eta \in \mathcal{P}(X)$ we make a change of variables $g = -\exp(h - 1)$, $h \in C(X)$ to obtain

$$F^*(\eta) = \sup_{h \in C(X)} \{\int h dP - \int e^{h-1} d\eta\} - 1 = R(P\|\eta) - 1. \tag{96}$$

Here we used the Legendre-transform variational representation of the KL divergence; see equation (1) in Birrell et al. (2022c) with $f(x) = x \log(x)$. Combining these results we obtain

$$\inf_{\eta \in \mathcal{P}(X)} \{R(P\|\eta) + W^\Gamma(Q, \eta)\} \ge \lim_{\alpha \to 1^-} R_\alpha^{\Gamma, IC}(P\|Q) \tag{97}$$

$$\ge \sup_{g \in \Gamma: g < 0} \left\{ \int g dQ + \int \log|g| dP \right\} + 1$$

$$= \inf_{\eta \in M(X)} \{F^*(\eta) + W^\Gamma(Q, \eta)\} + 1$$

$$= \inf_{\eta \in \mathcal{P}(X)} \{R(P\|\eta) + W^\Gamma(Q, \eta)\}.$$

This completes the proof. □

Now we prove Theorem 4.5, regarding the $\alpha \to \infty$ limit of the IC-$\Gamma$-Rényi divergences.

**Theorem B.7.** *Let $\Gamma \subset C(X)$ be admissible and $P, Q \in \mathcal{P}(X)$. Then*

$$\lim_{\alpha \to \infty} \alpha R_\alpha^{\Gamma/\alpha, IC}(P\|Q) = \inf_{\eta \in P(X)} \{D_\infty(P\|\eta) + W^\Gamma(Q, \eta)\} \tag{98}$$

$$= \sup_{g \in \Gamma: g < 0} \left\{ \int g dQ + \log \int |g| dP \right\} + 1. \tag{99}$$

*Proof.* First note that

$$\alpha R_\alpha^{\Gamma/\alpha, IC}(P\|Q) = \inf_{\eta \in \mathcal{P}(X)} \{\alpha R_\alpha(P\|\eta) + W^\Gamma(Q, \eta)\} \tag{100}$$

is nondecreasing in $\alpha$, therefore for $\eta \in \mathcal{P}(X)$ we have

$$\lim_{\alpha \to \infty} \alpha R_\alpha^{\Gamma/\alpha, IC}(P\|Q) = \sup_{\alpha > 1} \alpha R_\alpha^{\Gamma/\alpha, IC}(P\|Q) \tag{101}$$

$$\le \sup_{\alpha > 1} \{\alpha R_\alpha(P\|\eta) + W^\Gamma(Q, \eta)\}$$

$$= D_\infty(P\|\eta) + W^\Gamma(Q, \eta).$$

Maximizing over $\eta$ gives the upper bound

$$\lim_{\alpha \to \infty} \alpha R_\alpha^{\Gamma/\alpha, IC}(P\|Q) \le \inf_{\eta \in \mathcal{P}(X)} \{D_\infty(P\|\eta) + W^\Gamma(Q, \eta)\}. \tag{102}$$

To prove the reverse inequality, use the variational formula (53) to write

$$\alpha R_\alpha^{\Gamma/\alpha, IC}(P\|Q) = \alpha \sup_{g\in\Gamma: g<0}\left\{\int g/\alpha dQ + \frac{1}{\alpha-1}\log\int |g/\alpha|^{(\alpha-1)/\alpha}dP\right\} + \log\alpha + 1 \quad (103)$$

$$= \sup_{g\in\Gamma: g<0}\left\{\int gdQ + \frac{\alpha}{\alpha-1}\log\int |g|^{(\alpha-1)/\alpha}dP\right\} + 1.$$

Therefore, for all $g \in \Gamma$, $g < 0$ we can use the dominated convergence theorem to compute

$$\lim_{\alpha\to\infty}\alpha R_\alpha^{\Gamma/\alpha, IC}(P\|Q) \geq \int gdQ + \lim_{\alpha\to\infty}\frac{\alpha}{\alpha-1}\log\int |g|^{(\alpha-1)/\alpha}dP + 1 \quad (104)$$

$$= \int gdQ + \log\int |g|dP + 1.$$

Maximizing over $g$ then gives

$$\lim_{\alpha\to\infty}\alpha R_\alpha^{\Gamma/\alpha, IC}(P\|Q) \geq \sup_{g\in\Gamma: g<0}\left\{\int gdQ + \log\int |g|dP\right\} + 1. \quad (105)$$

Next we use the Fenchel-Rockafellar duality to derive a dual formulation of the right hand side of (105). Define $G, F : C(X) \to (-\infty, \infty]$, $G[g] = \infty 1_{g\notin\Gamma} - E_Q[g]$, $F[g] = \infty 1_{g\not<0} - \log\int |g|dP$. It is straightforward to prove that $G, F$ are convex and $G[-1] < \infty$, $F[-1] < \infty$ and $F$ is continuous at $-1$. Therefore Fenchel-Rockafellar duality implies

$$\inf_{g\in C(X)}\{F[g] + G[g]\} = \sup_{\eta\in C(X)^*}\{-F^*[-\eta] - G^*[\eta]\}, \quad (106)$$

i.e.

$$\sup_{g\in\Gamma: g<0}\left\{E_Q[g] + \log\int |g|dP\right\} = \inf_{\eta\in M(X)}\{F^*[\eta] + W^\Gamma(Q, \eta)\}, \quad (107)$$

where $F^*[\eta] = \sup_{g\in C(X): g<0}\{\int gd\eta + \log\int |g|dP\}$. We now prove that the infimum over $M(X)$ can be restricted to $\mathcal{P}(X)$. First suppose $\eta(X) \neq 1$. Then, because $\Gamma$ contains the constant functions, we have

$$W^\Gamma(Q, \eta) \geq \pm n(1 - \eta(X)) \to \infty \quad (108)$$

as $n \to \infty$ for appropriate choice of sign. Therefore $W^\Gamma(Q, \eta) = \infty$ when $\eta(X) \neq 1$.

Now suppose $\eta \in M(X)$ is not positive. Take a measurable set $A$ with $\eta(A) < 0$. By Lusin's theorem, for all $\epsilon > 0$ there exists a closed set $E_\epsilon \subset X$ and a continuous function $g_\epsilon \in C(X)$ such that $|\eta|(E_\epsilon^c) < \epsilon$, $0 \leq g_\epsilon \leq 1$, and $g_\epsilon|_{E_\epsilon} = 1_A$. Define $g_{n,\epsilon} = -ng_\epsilon - 1$, $n \in \mathbb{Z}^+$. Then $g_{n,\epsilon} \in \{g \in C(X) : g < 0\}$, hence

$$F^*[\eta] \geq \int g_{n,\epsilon}d\eta + \log\int |g_{n,\epsilon}|dP \quad (109)$$

$$= n(|\eta(A)| - \int (g_\epsilon - 1_A)1_{E_\epsilon^c}d\eta) - \eta(X) + \log(n\int g_\epsilon dP + 1)$$

$$\geq n(|\eta(A)| - \epsilon) - \eta(X).$$

Letting $\epsilon = |\eta(A)|/2$ and then taking $n \to \infty$ we see that $F^*[\eta] = \infty$ when $\eta$ is not positive. Together these results imply

$$\inf_{\eta\in M(X)}\{F^*[\eta] + W^\Gamma(Q, \eta)\} = \inf_{\eta\in\mathcal{P}(X)}\{F^*[\eta] + W^\Gamma(Q, \eta)\}. \quad (110)$$

Finally, using Theorem A.5 we see that

$$F^*[\eta] + 1 = \sup_{g\in C(X): g<0}\{\int gd\eta + \log\int |g|dP\} + 1 = D_\infty(P\|\eta) \quad (111)$$

for all $\eta \in \mathcal{P}(X)$. Combining these results gives

$$
\begin{aligned}
\lim_{\alpha \to \infty} \alpha R_\alpha^{\Gamma/\alpha, IC}(P\|Q) &\geq \sup_{g \in \Gamma : g < 0} \left\{ \int g dQ + \log \int |g| dP \right\} + 1 \qquad (112) \\
&= \inf_{\eta \in M(X)} \{ F^*[\eta] + W^\Gamma(Q, \eta) \} + 1 \\
&= \inf_{\eta \in \mathcal{P}(X)} \{ D_\infty(P\|\eta) + W^\Gamma(Q, \eta) \} \geq \lim_{\alpha \to \infty} \alpha R_\alpha^{\Gamma/\alpha, IC}(P\|Q) \,.
\end{aligned}
$$

This completes the proof. $\qquad \square$

Finally, we prove Theorem 3.5, the data-processing inequality for the IC-$\Gamma$-Rényi divergences. First we introduce the following notation: Let $Y$ be another compact metric space and $K$ be a probability kernel from $X$ to $Y$. Given $P \in \mathcal{P}(X)$ we denote the composition of $P$ with $K$ by $P \otimes K$ (a probability measure on $X \times Y$) and we denote the marginal distribution on $Y$ by $K[P]$. Given $g \in C(X \times Y)$ we let $K[g]$ denote the function on $X$ given by $x \mapsto \int g(x, y) K_x(dy)$.

**Theorem B.8** (Data Processing Inequality). *Let $\alpha \in (0, 1) \cup (1, \infty)$, $Q, P \in \mathcal{P}(X)$, and $K$ be a probability kernel from $X$ to $Y$ such that $K[g] \in C(X)$ for all $g \in C(X, Y)$.*

1. *If $\Gamma \subset C(Y)$ is admissible then*

$$
R_\alpha^{\Gamma, IC}(K[P]\|K[Q]) \leq R_\alpha^{K[\Gamma], IC}(P\|Q) \,. \qquad (113)
$$

2. *If $\Gamma \subset C(X \times Y)$ is admissible then*

$$
R_\alpha^{\Gamma, IC}(P \otimes K\|Q \otimes K) \leq R_\alpha^{K[\Gamma], IC}(P\|Q) \,. \qquad (114)
$$

*Proof.* It is straightforward to show that admissiblility of $\Gamma$ implies admissibility of $K[\Gamma]$. Hence we can write

$$
R_\alpha^{K[\Gamma], IC}(P\|Q) = \sup_{\tilde{g} \in K[\Gamma] : \tilde{g} < 0} \left\{ \int \tilde{g} dQ + \frac{1}{\alpha - 1} \log \int |\tilde{g}|^{(\alpha-1)/\alpha} dP \right\} + \alpha^{-1}(\log \alpha + 1) \quad (115)
$$

$$
\geq \sup_{g \in \Gamma : g < 0} \left\{ \int K[g] dQ + \frac{1}{\alpha - 1} \log \int |K[g]|^{(\alpha-1)/\alpha} dP \right\} + \alpha^{-1}(\log \alpha + 1) \,.
$$

Using Jensen's inequality we can derive

$$
|\int g(y) K_x(dy)|^{(\alpha-1)/\alpha} \leq \int |g(y)|^{(\alpha-1)/\alpha} K_x(dy) \,, \quad \alpha \in (0, 1) \,, \qquad (116)
$$

$$
|\int g(y) K_x(dy)|^{(\alpha-1)/\alpha} \geq \int |g(y)|^{(\alpha-1)/\alpha} K_x(dy) \,, \quad \alpha > 1 \,. \qquad (117)
$$

Combining (115) - (117) with the monotonicity of $y \mapsto \frac{1}{\alpha-1} \log(y)$ we arrive at (113). The proof of (114) is similar. $\qquad \square$

## C  DETAILS ON ANALYTICAL EXAMPLES AND COUNTEREXAMPLES

In this appendix we present several details regarding the analytical examples found in Section 5.

### C.1  INFIMAL CONVOLUTION AND SCALING LIMITS

First we present a simple example that illustrates the infimal convolution formula and limiting properties from Sections 3 and 4.

Let $P = \delta_0$, $Q_{x,c} = c\delta_0 + (1-c)\delta_x$ for $c \in (0, 1)$, $x > 0$, and let $\Gamma = \text{Lip}^1$. Then for $L > 0$ one can compute

$$
R_\alpha(P\|Q_{x,c}) = \alpha^{-1} \log(1/c) \,, \qquad (118)
$$

$$
W^{L\Gamma}(Q_{x,c}, P) = (1-c)Lx \,, \qquad (119)
$$

and

$$R_\alpha^{L\Gamma,IC}(P\|Q_{x,c}) = \sup_{a,b<0:|a-b|\leq x} \{Lca + L(1-c)b + \alpha^{-1}\log(L|a|)\} + \alpha^{-1}(\log\alpha + 1) \quad (120)$$

$$= \sup_{a<0}\{Lca + L(1-c)\min\{x+a,0\} + \alpha^{-1}\log(L|a|)\} + \alpha^{-1}(\log\alpha + 1)$$

$$= \alpha^{-1} + \alpha^{-1}\sup_{y>0}\begin{cases} -cy + \log y\,, & y \leq \alpha Lx \\ (1-c)\alpha Lx - y + \log y\,, & y > \alpha Lx \end{cases}$$

$$= \begin{cases} (1-c)Lx\,, & 0 < \alpha Lx < 1 \\ \alpha^{-1} - cLx + \alpha^{-1}\log(\alpha Lx)\,, & 1 \leq \alpha Lx \leq 1/c \\ \alpha^{-1}\log(1/c)\,, & \alpha Lx > 1/c \end{cases} \quad .$$

In particular, it is straightforward to show that

$$R_\alpha^{L\Gamma,IC}(P\|Q_{x,c}) \leq W^{L\Gamma}(Q_{x,c}, P)\,, \quad (121)$$

$$\lim_{x\to 0^+} R_\alpha^{L\Gamma,IC}(P\|Q_{x,c}) = \lim_{x\to 0^+}(1-c)Lx = 0\,, \quad (122)$$

$$\lim_{L\to\infty} R_\alpha^{L\Gamma,IC}(P\|Q_{x,c}) = \alpha\log(1/c) = R_\alpha(P\|Q_{x,c})\,. \quad (123)$$

We can also rewrite this in terms of the solution to the infimal convolution problem as follows

$$R_\alpha^{L\Gamma,IC}(P\|Q_{x,c}) = \begin{cases} W^{L\Gamma}(Q_{x,c}, P)\,, & 0 < \alpha Lx < 1 \\ R_\alpha(P\|Q_{x,1/(\alpha Lx)}) + W^{L\Gamma}(Q_{x,c}, Q_{x,1/(\alpha Lx)})\,, & 1 \leq \alpha Lx \leq 1/c \\ R_\alpha(P\|Q_{x,c})\,, & \alpha Lx > 1/c \end{cases} \quad .$$

$$(124)$$

Taking the worst-case-regret scaling limit we find

$$\lim_{\alpha\to\infty} \alpha R_\alpha^{\Gamma/\alpha,IC}(P\|Q_{x,c}) = \begin{cases} (1-c)x\,, & 0 < x < 1 \\ 1 - cx + \log(x)\,, & 1 \leq x \leq 1/c \\ \log(1/c)\,, & x > 1/c \end{cases} \quad (125)$$

$$= \begin{cases} W^\Gamma(Q_{x,c}, P)\,, & 0 < x < 1 \\ D_\infty(P\|Q_{x,1/x}) + W^\Gamma(Q_{x,c}, Q_{x,1/x})\,, & 1 \leq x \leq 1/c \\ D_\infty(P\|Q_{x,c})\,, & x > 1/c \end{cases} \quad ,$$

where $D_\infty(P\|Q_{x,c}) = \log(1/c)$.

## C.2 Γ-RÉNYI-DONSKER-VARADHAN COUNTEREXAMPLE

As an alternative to Definition 3.1, one can attempt to regularize the Rényi divergences by restricting the test-function space in the variational representation (3), leading to the Γ-**Rényi-Donsker-Varadhan** divergences

$$R_\alpha^{\Gamma,DV}(P\|Q) := \sup_{\phi\in\Gamma}\left\{\frac{1}{\alpha-1}\log\int e^{(\alpha-1)\phi}dP - \frac{1}{\alpha}\log\int e^{\alpha\phi}dQ\right\}\,. \quad (126)$$

The bound

$$\log\int e^{c\phi}dP \geq c\int\phi dP\,, \quad \phi\in\Gamma, c\in\mathbb{R} \quad (127)$$

implies that $R_\alpha^{\Gamma,DV} \leq W^\Gamma$ for $\alpha \in (0,1)$, making (126) a useful regularization of the Rényi divergences in this case; this utility was demonstrated in Pantazis et al. (2022), where it was used to construct GANs. However, the representation (126) is known to be poorly behaved when $\alpha > 1$. Here we provide a counterexample showing that, unlike for the IC-Γ-Rényi divergences, $R_\alpha^{\Gamma,DV} \not\leq W^\Gamma$ in general when $\alpha > 1$. We conjecture that this fact is the reason for the poor behavior of $R_\alpha^{\Gamma,DV}$ when $\alpha > 1$.

Let $P_{x,c} = c\delta_0 + (1-c)\delta_x$, $Q = \delta_0$ for $x > 0$, $c \in (0,1)$ and $\Gamma_L = \text{Lip}^L$. Then for $\alpha > 1$ we have

$$R_\alpha^{\Gamma_L,DV}(P_{x,c}\|Q) = \sup_{a,b\in\mathbb{R}:|a-b|\leq Lx} \left\{ \frac{1}{\alpha-1} \log(c\exp((\alpha-1)a) + (1-c)\exp((\alpha-1)b)) - a \right\} \tag{128}$$

$$= \sup_{a\in\mathbb{R}} \left\{ \frac{1}{\alpha-1} \log(c\exp((\alpha-1)a) + (1-c)\exp((\alpha-1)(Lx+a))) - a \right\}$$

$$= \frac{1}{\alpha-1} \log\left(c + (1-c)\exp((\alpha-1)Lx)\right),$$

$$W^{\Gamma_L}(P_{x,c},Q) = \sup_{|a-b|\leq Lx} \{ca + (1-c)b - a\} = (1-c)Lx. \tag{129}$$

Note that the condition $\alpha > 1$ was crucial in computing the supremum over $b$ in (128). Using strict concavity of the logarithm one can then obtain the bound

$$R_\alpha^{\Gamma_L,DV}(P_{x,c}\|Q) > W^{\Gamma_L}(P_{x,c},Q). \tag{130}$$

## C.3 log-$\Gamma$-RÉNYI-DONSKER-VARADHAN COUNTEREXAMPLE

A second alternative to Definition 3.1 is to again start with (3) and then reduce the test-function space to $\frac{1}{\alpha}\log(\Gamma)$

$$R_\alpha^{\Gamma,log-DV}(P\|Q) := \sup_{g\in\Gamma:g>0} \left\{ \frac{1}{\alpha-1} \log \int g^{(\alpha-1)/\alpha} dP - \frac{1}{\alpha} \log \int g\, dQ \right\}. \tag{131}$$

However, as we show below, this definition fails to provide a regularized divergence; in particular, it is incapable of meaningfully comparing Dirac distributions.

Let $P = \delta_0$, $Q_x = \delta_x$, $x > 0$, $\Gamma_L = \text{Lip}^L$. Then straightforward computations using the variational definition gives

$$R_\alpha^{\Gamma_L,DV-log}(P\|Q_x) = \alpha^{-1} \sup_{\phi\in\Gamma,\phi>0} \log(\phi(0)/\phi(x)) \tag{132}$$

$$= \alpha^{-1} \sup_{b>0} \sup_{a>0:b-x\leq a\leq x+b} \log(a/b)$$

$$= \alpha^{-1} \sup_{b>0} \log(1 + x/b) = \infty.$$

In contrast we have

$$R_\alpha^{\Gamma_L,IC}(P\|Q_x) \tag{133}$$

$$= \sup_{a<0,b<0:b-x\leq a\leq b+x} \{Lb + \alpha^{-1}\log L + \alpha^{-1}\log(|a|)\} + \alpha^{-1}(\log(\alpha) + 1)$$

$$= \sup_{b<0} \{Lb + \frac{1}{\alpha}\log(|b-x|)\} + \alpha^{-1}\log L + \alpha^{-1}(\log\alpha + 1)$$

$$= \begin{cases} \alpha^{-1}\log(\alpha Lx) + \alpha^{-1}, & x \geq 1/(\alpha L) \\ Lx, & x < 1/(\alpha L) \end{cases}.$$

In particular,

$$R_\alpha^{\Gamma_L,IC}(P\|Q_x) \leq Lx = W^{\Gamma_L}(P,Q_x), \tag{134}$$

$$\lim_{x\to 0^+} R_\alpha^{\Gamma_L,IC}(P\|Q_x) = 0, \tag{135}$$

showing that $R_\alpha^{\Gamma_L,IC}$ is able to capture the convergence of $Q_x$ to $P$ as $x \to 0^+$, while $R_\alpha^{\Gamma,log-DV}$ fails to do so.

# D    ADDITIONAL EXAMPLES

## D.1    TRAINING SYMMETRY-PRESERVING GANS ON ROTMNIST

When learning a distribution $P$ that is invariant under a symmetry group (e.g., rotation invariance for images without preferred orientation) one can greatly increase performance by using a GAN that incorporates the symmetry information into the generator and the discriminator space $\Gamma$ Dey et al. (2021). A theory of such symmetry-preserving GANs was developed in Birrell et al. (2022b) and the new divergences introduced in this paper satisfy the assumptions required to apply that theory. In Table 2 we demonstrate this effectiveness on the RotMNIST dataset, obtained from randomly rotating the original MNIST digit dataset LeCun et al. (1998), resulting in an rotation-invariant distribution. Note that incorporating more symmetry information into the GAN (i.e., progressing down the rows of the table) results in greatly improved performance, especially in the low data regime.

Table 2: The median of the FIDs (lower is better), calculated every 1,000 generator update for 20,000 iterations, over three independent trials. The number of the training samples used for experiments varies from 1% (600) to 10% (6,000) of the RotMNIST training set. The NN structure and hyperparameters are the same as those used in Section 5.4 of Birrell et al. (2022b). `Eqv G` (resp. `Inv D`) denotes that the symmetry information was incorporated into the generator (resp. discriminator) while `CNN` implies that a convolutional NN was used (without rotational symmetry). $\Sigma$ denotes the rotation group used, where $C_n$ denotes rotations by being integer multiples of $2\pi/n$.

| | Architecture | 1% | 5% | 10% |
|---|---|---|---|---|
| | CNN G&D | 357 | 325 | 298 |
| | Eqv G + CNN D, $\Sigma = C_4$ | 464 | 271 | 263 |
| Reverse $R_2^{\Gamma, IC}$ | CNN G + Inv D, $\Sigma = C_4$ | 366 | 321 | 302 |
| | Eqv G + Inv D, $\Sigma = C_4$ | 151 | 105 | 89 |
| | Eqv G + Inv D, $\Sigma = C_8$ | **114** | **71** | **62** |

## D.2    VARIANCE OF RÉNYI ESTIMATORS

Here we compare the DV-Rényi and CC-Rényi estimators on the Gaussian test problem from Section 6.1, except in lower dimensions (1-D and 100-D). Qualitatively, the behavior is similar. In particular, it is striking that the DV-Rényi estimator performs extremely poorly even in the 1-D case (see Figure 3) while the CC-Rényi estimator has much lower variance and MSE when the separation between the distributions becomes larger (i.e., as $\mu_q$ increases).

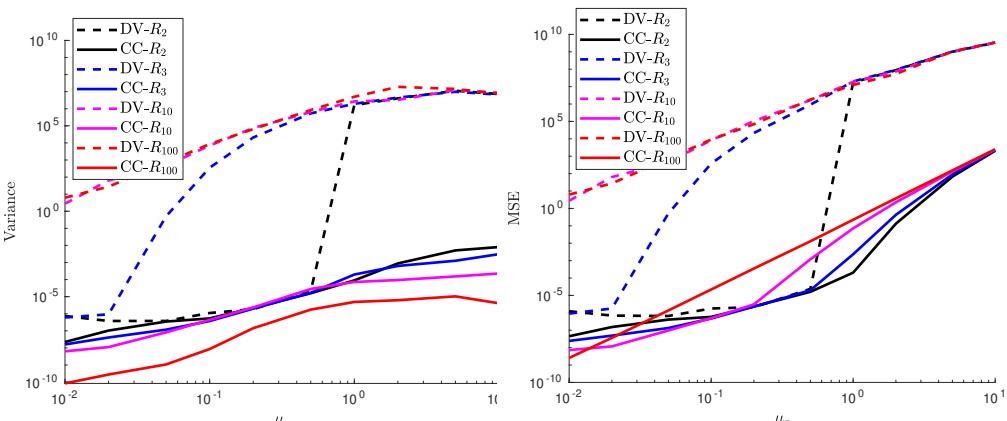

Figure 3: Variance and MSE of estimators of the classical Rényi divergence between 1-dimensional Gaussians. DV-$R_\alpha$ refers to Rényi divergence estimators built using (3) while CC-$R_\alpha$ refers to estimators built using our new variational representation (4). We used a NN with one fully connected layer of 64 nodes, ReLU activations, and a poly-softplus final layer (for CC-Rényi). We trained for 10000 epochs with a minibatch size of 500. The variance and MSE were computing using data from 50 independent runs. Note that the CC-Rényi estimator has significantly reduced variance and MSE compared to the DV-Rényi estimator, even when $\alpha$ is large.

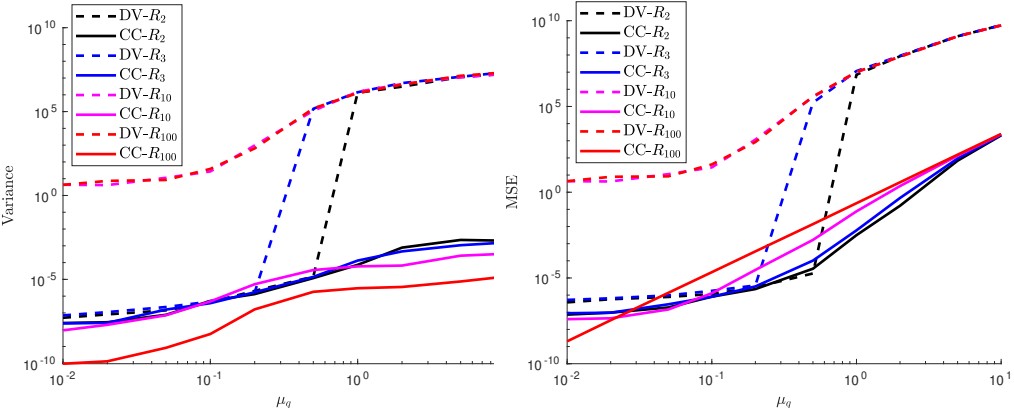

Figure 4: Variance and MSE of estimators of the classical Rényi divergence between 100-dimensional Gaussians. DV-$R_\alpha$ refers to Rényi divergence estimators built using (3) while CC-$R_\alpha$ refers to estimators built using our new variational representation (4). We used a NN with one fully connected layer of 64 nodes, ReLU activations, and a poly-softplus final layer (for CC-Rényi). We trained for 10000 epochs with a minibatch size of 500. The variance and MSE were computing using data from 50 independent runs. Again, the CC-Rényi estimator has significantly reduced variance and MSE compared to the DV-Rényi estimator, even when $\alpha$ is large.

