# OpenReview forum: "Function-space regularized Rényi divergences"
_ICLR.cc/2023/Conference — ICLR 2023 poster_

### Official Review · Reviewer_Fzvh · 2022-10-21

**Confidence:** 3
**Correctness:** 3
**Technical Novelty And Significance:** 3
**Empirical Novelty And Significance:** 3
**Recommendation:** 8

**Clarity, Quality, Novelty And Reproducibility:**

Overall, the paper was clearly written (bar a few of the comments listed in the above section). The quality and clarity are good. In terms of originality, the main results in this paper builds on the work within the literature and provides interesting new results.

**Details Of Ethics Concerns:**

N/A.

**Strength And Weaknesses:**

Overall the paper is nicely written and describes well Renyi divergence and its background, and provides a good description of the paper's main results. In particular, the paper provides an interesting development to the study of Renyi divergences by introducing a new family of regularized Renyi divergences.

Some minor comments on the strengths of the paper are:
1. The paper includes clear description of definitions and main results. The highlights of contributions on Page 2 were very helpful to understanding the paper.
2. The paper addresses Renyi divergences, an interesting and important topic, and provides interesting development to available results. The variational formulations of the Renyi divergence and the corresponding regularizations using an IPM (i.e., (6)) are very interesting.
3. The paper considers a variety of numerical experiments, demonstrating flexibility in applications.

Some minor comments on the weaknesses of the paper are:
1. In the first experiment (Section 6.1), it would be interesting to see if CC-Renyi also performs better than DV-Renyi when distributions are close. In the experiment shown the two Gaussians are chosen to be N(2,0.5) and N(0,1) respectively and relative variance and MSE numbers are quite extreme for DV-Renyi -- it would be interesting if CC-Renyi also improves upon DV-Renyi for scenarios where say the two Gaussians are N(0,1) and N(0.1,1), for example.
2. The first section in Section 5 may benefit from some more explanation.
3. It may be helpful to adjust some of the citation formatting to make the paper more readable. For example, in the very first sentence of the Introduction, the two citations are a bit confusing to the overall readability of the sentence.
4. The details to the last experiment seem to be slightly lacking; it would be helpful to add some more description to the main text.

**Summary Of The Paper:**

The paper discusses a regularized family of Renyi divergences using a variational function form. For this family of divergences the paper discusses various properties relating to limits, interpolations, data processing inequality. Examples of the importance of using the new variational Renyi divergences is demonstrated through examples (and counterexamples). Finally, the paper ends with a discussion of numerical applications: a) the variance of the convex-conjugate Renyi divergence estimators are much lower than that of estimators designed using the Renyi-Donsker-Varadhan variational formula when risk sensitive terms are detrimental to the DV-Renyi estimators; b) proposed infimal-convolution divergences are more suitable in the detection of rare sub-populations for a particular dataset; c) the proposed infimal-convolution divergences can perform better than Wasserstein GAN on the CIFAR-10 dataset; and d) the proposed divergences provides improvement on the ROTMNIST dataset.

**Summary Of The Review:**

Below is a short summary of justification, for more details please refer to the above sections.

1. The paper is clearly written and states the background and main results adequately. Main ideas such as the addition of regularization to Renyi divergences are nicely conveyed.
2. The paper introduces interesting new results for the important field of Renyi divergences, and is closely related to the scope of ICLR.
3. Overall, no major issues were identified in the main text.

---

> ### Author Response · Authors · 2022-11-18
> **Response to Reviewer Fzvh**
>
> 1) In the first experiment (Section 6.1), it would be interesting to see if CC-Renyi also performs better than DV-Renyi
> when distributions are close. In the experiment shown the two Gaussians are chosen to be N(2,0.5) and N(0,1) respectively and relative variance and MSE numbers are quite extreme for DV-Renyi -- it would be interesting if CC-
> Renyi also improves upon DV-Renyi for scenarios where say the two Gaussians are N(0,1) and N(0.1,1), for example.
>
> Response:  We have modified this example to show the variance and MSE over a range of values of the mean (see Figure 1).  While the performance of DV-Renyi does improve when the difference in means is smaller, we still find that CC-Renyi performs significantly better. The original example and table has been moved to the appendix.
>
> We also refer to very recent theoretical work on lower bounds of the variance for the DV-Renyi in Theorem 4.1 of Lee & Shin (2022).
>
> 2) The first section in Section 5 may benefit from some more explanation.
>
> Response: The examples in the first part of Section 5 were only meant to present the simplest possible example that illustrates the theoretical properties from Theorems 3.4, 4.1, and 4.5.
>
> 3) It may be helpful to adjust some of the citation formatting to make the paper more readable. For example, in the very first sentence of the Introduction, the two citations are a bit confusing to the overall readability of the sentence.
>
> Response: We are using here the format of ICLR, but we have made some changes to improve readability.
>
> 4) The details to the last experiment seem to be slightly lacking; it would be helpful to add some more description to
> the main text.
>
> Response: We moved the rotMNIST example to the appendix due to space constraints and so we omit implementation details; these  can be found in  Birrell et al. (2022b).

---

> > ### Comment · Reviewer_Fzvh · 2022-11-24
> > **Response to rebuttal**
> >
> > Thanks for the responses. I recommend careful revisions to future versions; for example, the first paragraph of Section 5 is lacking a period.
> >
> > An additional comment on bibliography and citations: it might be possible to change some of the citation format from NAME, (YEAR) to (NAME, YEAR), see, for example, commands \citet{} and \citep{} also available in the ICLR style template.

---

### Official Review · Reviewer_ht2a · 2022-10-25

**Confidence:** 3
**Clarity, Quality, Novelty And Reproducibility:** The paper is clear, well-written and …
**Correctness:** 4
**Technical Novelty And Significance:** 3
**Empirical Novelty And Significance:** 2
**Recommendation:** 5

**Strength And Weaknesses:**

Strength:

1. The paper proposes a new divergence which include the Renyi divergence and other IPM. It is new and have some interesting properties.
2. The author derives some properties of the new divergence and show its dual form for computation.
3. They derive the limit formula of divergence and present some special case of the new divergence.
4. Empirical results show that the new divergence is more stable the Renyi divergence when $\alpha >1$.

Weaknesses:

1. The derivations of mathematical results of the paper are quite straightforward from known results.
2. It is not clear for me the popularity of using new divergence in reality when $\alpha >1$. For $\alpha <1$, I do not see any advantage to have new divergence.
3. In the application on biological datasets, the best results are obtained when $\alpha = \infty$. It means that we do not need to care about other values of $\alpha \in (1,\infty)$, then why do we need to define the new divergence.
4. In the experiment with GAN, the proposed divergence is used to improve the performance of generative model. The results are shown through the FID scores. We need to choose a particular $\alpha$ such that it obtains the lowest FID. In this case, is there any explanation of using the proposed divergence rather than searching something in a larger space will obtain a lower minimum.



**Summary Of The Paper:**

The paper proposes a new divergence which is a convolution between Renyi divergence and other IPM.


**Summary Of The Review:**

A new divergence is introduced, some its properties are explored, however the real need of new divergence for applications is not well justified.

---

> ### Author Response · Authors · 2022-11-18
> **Response to Reviewer ht2a**
>
> 1) The derivations of mathematical results of the paper are quite straightforward from known results.
>
> Response:
> In fact the derivations here are elaborate and new in the context of Renyi divergences. On one hand, the very recent article Birrell et al. (2022a) applies to the class of divergences known as $f$-divergences. However,  Renyi divergences (and worst-case-regret) are not $f$-divergences and so the results of Birrell et al. (2022a) do not apply here.  In this paper we adapted the infimal-convolution method of Birrell et al. (2022a)   to the Renyi divergence setting, which required several technical innovations, including the development of a new variational formula for the classical Renyi divergences and worst-case-regret, Theorems 2.1 and 2.2; attempts based on the previously known Renyi-Donsker-Varadhan variational formula fail as discussed in Section 5. We have emphasized this point at the end of the related work section on page 2. Furthermore the proposed worst case regret (WCR) variational formula and estimator are entirely new and are obtained from  the regularized Rényi divergence in the  $\alpha \to \infty$ asymptotic limit.
>
>
>
> 2)  It is not clear for me the popularity of using new divergence in reality when $\alpha>1$. For $\alpha<1$, I do not see any
> advantage to have new divergence.
>
> Response: Thank you for the opportunity to clarify this important point. Part of this response has also been included in the paper:
>
>
>   The order of Renyi divergence controls the weight put on the tails, with the limiting cases being mode-covering and mode-selection Minka (2005). Renyi divergence estimation is used in a number of applications, including Sajid et al. (2022) (behavioural sciences),  Mironov (2017)  (differential privacy), and  Li & Turner (2016) (variational inference);  in the latter the variational formula is an adaptation of the evidence lower bound. Renyi divergences have been also applied in the training of GANs  Bhatia et al. (2020)  (loss function for binary classification - discrete case) and in   Pantazis et al. (2022) (continuous case, based on the Renyi-Donsker-Varahdan variational formula).   Renyi divergences with  $\alpha >1$ are also used in contrastive representation learning, Lee
> & Shin (2022),  as well as in PAC-Bayesian Bounds,  Begin et al. (2016). In the context of uncertainty quantification and sensitivity analysis, Renyi divergences provide confidence bounds for rare events, Atar et al. (2015); Dupuis et al. (2020), with higher rarity corresponding to larger $\alpha$.
>
>  However, we also note that the lack of low variance estimators for Renyi divergences has prevented wide-spread and accessible experimentation with this class of information-theoretic tools, except in very special cases. We hope our results here  will provide a suitable set of tools to address this gap in the methodology.
>
>
>
> 3) In the application on biological datasets, the best results are obtained when $\alpha=\infty$. It means that we do not need
> to care about other values of $\alpha\in(1,\infty)$, then why do we need to define the new divergence.
>
> Response:
> a. The derivation of the regularized WCR divergence requires the definition of the regularized Renyi divergence and the passing to the $\alpha \to \infty$ limit. In this sense, it is necessary to define the new divergence for $\alpha < \infty$ first in order to get the optimal result.
>
> b. Furthermore, as discussed in the response to the second  comment above, low variance estimators for the $\alpha$-Renyi divergence for finite $\alpha$'s are also of independent interest in numerous other application domains.
>
> 4) In the experiment with GAN, the proposed divergence is used to improve the performance of generative model. The
> results are shown through the FID scores. We need to choose a particular such that it obtains the lowest FID. In
> this case, is there any explanation of using the proposed divergence rather than searching something in a larger
> space will obtain a lower minimum.
>
> Response:
> Indeed the referee is correct in the sense that in GANs there is a tremendous (in fact infinite)  flexibility
> to select divergences of probability metrics for adversarial training.  We selected the Renyi family to experiment with because in its regularized form proposed here it  includes some well-studied GANs as pointed out in   Pantazis et al. (2022). Namely, Wasserstein GAN, MS-GAN (based on $\chi^2$ which is Renyi with $\alpha=2$), KL,  Reverse KL and Hellinger-based  GANs.

---

### Official Review · Reviewer_qiWv · 2022-10-26

**Confidence:** 3
**Correctness:** 3
**Technical Novelty And Significance:** 3
**Empirical Novelty And Significance:** 3
**Recommendation:** 6

**Clarity, Quality, Novelty And Reproducibility:**

It is not easy to follow the ideas in the paper, some more explanations may be useful to improve the flows of the paper (i.e., some notations are given without definitions/explanations which makes readers hard to follow the proposed ideas).
Overall, I think the proposed divergences and their properties are important. However, the authors may need to elaborate with more details to improve the flows of the papers and highlight the new contributions of the submission (and distinguish with results in the literature)

Some concerns are as follows:
+ Equ. (1) is for Renyi divergence and Equ. (2) is for Renyi-Donsker-Varadhan variational formula. Are these equations equivalent? (If not, the authors should use different notations for the Renyi divergence and its variational formula). It is better in case the authors elaborate the relations between (1) and (2)
+ In Equ. (1), the authors should describe the meaning of the notation (P << Q)
+ It seems that the definitions of infimal-convoluation \Gamma-Renyi divergence appears in Birrell et al, JMLR'2022a. Could the authors elaborate the relation between these divergences. Are they the same? or what are the key differences between them. It is also better in case the authors distinguish the relation between the submission and Birrell et al, JMLR'2022a, and emphasize the novelty of the submission.
+ How the authors choose the \alpha nd \Gamma space for the proposed divergence in applications? especially the \Gamma space? How does the choice of \Gamma affect the proposed method in experiments?
+ In Section 3, the authors give some important choices for \Gamma. Could the authors elaborate how to compute the proposed method for these important cases?

**Strength And Weaknesses:**

(*) Strength:
+ The authors propose a new family of regularized Renyi divergences by infimal-convolution with IPM.
+ The authors derive several theory results for the proposed divergence

(*) Weaknesses:
+ It seems that some results in the submission appear in the Birrell et al, JMLR'2022a.
+ It is better to give definitions/explanations for some notations before using them.
+ It is unclear how to choose the hyperparameters of the proposed divergences (it seems that these parameters play important roles in applications)

**Summary Of The Paper:**

The authors propose a new family of regularized Renyi divergences by infimal-convolution with integral probability metric (IPM). The authors derive its dual variational representation and provide several theoretical results about limits, interpolations, regularized worse-case regret. The authors evaluate the proposed divergence on both synthetic and real datasets, and its application on GANs.

**Summary Of The Review:**

The proposed divergences and their properties are interesting. However, the authors may need to elaborate with more details to highlight the contribution and distinguish with results in the literature (especially Birrell et al, JMLR'2022a)

---

> ### Author Response · Authors · 2022-11-17
> **Response to Reviewer qiWv**
>
> 1) Equ. (1) is for Renyi divergence and Equ. (2) is for Renyi-Donsker-Varadhan variational formula. Are these equations
> equivalent? (If not, the authors should use different notations for the Renyi divergence and its variational formula). It is better in case the authors elaborate the relations between (1) and (2)
>
> Response:
> Equation 2 (the old Eq. 1) is the definition of Renyi divergences, which we have emphasized by changing the equals sign to the equal-by-definition symbol ($\coloneqq$).  Eq 3. (the old Eq. 2) is a variational representation of $R_\alpha$  that was previously derived in the literature and which we use to derive our new results.  We have changed the wording above Eq. 3 to clarify this.
>
> 2) In Equ. (1), the authors should describe the meaning of the notation (P $\ll$ Q)
>
> Response: This notation denotes absolute continuity of P with respect to Q.  We have clarified this below equation 2.
>
>
> 3) It seems that the definitions of infimal-convoluation $\Gamma$-Renyi divergence appears in Birrell et al, JMLR'2022a.
> Could the authors elaborate the relation between these divergences. Are they the same? or what are the key
> differences between them. It is also better in case the authors distinguish the relation between the submission and
> Birrell et al, JMLR'2022a, and emphasize the novelty of the submission.
>
> Response:
> Birrell et al, JMLR'2022a applies to the class of divergences known as $f$-divergences.  Renyi divergences (and worst-case-regret) are not $f$-divergences and so the results of Birrell et al, JMLR'2022a do not apply here.  In this paper we adapted the infimal-convolution method of Birrell et al, JMLR'2022a  to the Renyi divergence setting, which required several technical innovations, including the development of a new variational formula for the classical Renyi divergences and worst-case-regret, Theorems 2.1 and 2.2; attempts based on the previously known Renyi-Donsker-Varadhan variational formula fail as discussed in Section 5. We have emphasized this point at the end of the related work section on page 2.
>
>
> 4) How the authors choose the $\alpha$ and $\Gamma$ space for the proposed divergence in applications? especially the
> $\Gamma$ space? How does the choice of $\Gamma$ affect the proposed method in experiments?
>
> Response:
>  Selecting  $\alpha$, and more generally a particular divergence, to
> obtain an optimal solution for a particular application is still an open question.
>  This is largely a trial and error process, although there are practical insights related to mode-seeking vs. mass-covering in Minka (2005); Li & Turner (2016) on how to choose divergences.  In the context of uncertainty quantification and sensitivity analysis of rare events,  some theoretical  understanding on  connections between specific quantities of interest (rare events)  and $\alpha$ can be found in  Atar et al. (2015); Dupuis et al. (2020).
>
>   The choice  of $\Gamma$ is mathematically  dictated by the choice of the $\Gamma$-IPM used in the definition in Eq. 1. For example, $\Gamma$ is the space of all 1-Lipschitz functions when the IPM is the   1-Wasserstein metric. Practically, the choice of $\Gamma$ imposes a regularization in the divergences. For instance, in Birrell et al. (2022b) structure-preserving GANs were introduced as a data-efficient framework for learning probability measures with embedded structures;  there efficient adversarial learning is achieved by reducing the discriminator space to its projection onto its invariant subspace $\Gamma$.
>
> 5) In Section 3, the authors give some important choices for $\Gamma$. Could the authors elaborate how to compute
> the proposed method for these important cases?
>
> Response:
> In practice, uniform bounds can be implemented using an appropriately chosen final NN layer. Lipschitz bounds can be implemented using spectral normalization of neural networks Miyato et al. (2018), or using a soft gradient penalty  Gulrajani et al. (2017). The function space   $\Gamma$ for structure-preserving GANs discussed in the Appendix is implemented using equivariant neural networks, Birrell et al. (2022b). If $\Gamma$ is a ball in an RKHS space the implementation is carried out  using the same tools used in e.g. MMD-type distances and divergences,  Gretton et al. (2012); Glaser et al. (2021). We have added a paragraph describing this to the paper.

---

> > ### Comment · Reviewer_qiWv · 2022-12-05
> > **Thank you for your explanation**
> >
> > Thank you for your explanation.
> > The updated version has better clarification over the raised points.
> > However, the submission shares a large part with the Birrell et al, JMLR'2022a; the extension to Renyi divergences (and worst-case-regret) is quite marginal given the work of Birrell et al, JMLR'2022a (although it is quite new in the literature).
> >
> > The interesting parts are the hyperparameters \alpha, \Gamma. However, for application perspective, choosing those hyperparameters look like to search bigger space (confirmed by explanations for Reviewer ht2a), which may limit the merit of the new proposed family of regularized Renyi divergences.
> >
> > Overall, I think the proposed ideas may be interesting. However, the hyperparameters are essentially important, but hard-to-be-chosen for applications (besides exhaustive search with a "larger" space). So, I keep my score and still lean on the acceptance.

---

### Comment · Area_Chair_NR2p · 2022-11-19
**Please respond to author feedback**

Dear reviewers,

The authors have provided their feedback. Please respond, and at least acknowledge you've read them.

Best, AC

---

### Decision · Program_Chairs · 2023-01-20

**Decision:**

Accept: poster

**Justification For Why Not Higher Score:**

There are concerns regarding whether the theoretical contribution is significantly different than Birrell et al. JMLR 2022 paper. Also the experiments on GANs do not show significant practical advantages of the associated GAN algorithm for the proposed divergence.

**Justification For Why Not Lower Score:**

I believe the ideas and results should be of interest to a variety of ML researchers working on GANs, variational inference and differential privacy, where Renyi divergence has been applied to applications in these fields.

**Metareview: Summary, Strengths And Weaknesses:**

The authors propose a new family of Renyi divergences by infimal-convolution with integral probability metric (IPM). The paper is mainly theoretical, with theorems on different aspects of the proposed divergence (e.g., limits, interpolations, regularised worse-case regret). Experiments considered applications to training GANs, with small improvements over baselines.

Reviewers overall welcomed the development of this new family of divergences and the analyses of their theoretical properties in multiple different aspects. The synthetic experiments on reporting variances of Renyi divergence estimates show significant advantages of the proposed approach. Given that high variance of Renyi divergence estimates is a major obstacle for practical usage, I consider these results as showing promising progress towards addressing this challenge.

However, the main concerns are (1) whether the theoretical development is significant as compared with prior work (Birrell et al. JMLR 2022), and (2) the experimental results on GANs do not show significant gains.

In revision, I suggest the authors to better clarify their theoretical contribution as compared with Birrell et al. JMLR 2022, and improve their experiments, e.g., testing variances of the estimators for higher dimensional distributions.

**Note From Pc:**

if the above contains the word "oral" or "spotlight" please see: "oral" presentation means -> notable-top-5% and "spotlight" means -> notable-top-25%. As stated in our emails, we are disassociating presentation type from AC recommendations

**Summary Of Ac-Reviewer Meeting:**

N/A